# Interaction hub critical for telomerase recruitment and primer-template handling for catalysis

Shilpa Padmanaban, Valerie M Tesmer, Jayakrishnan Nandakumar

**Telomerase processively adds telomeric DNA repeats to chromosome ends using catalytic protein subunit TERT and a template on its RNA subunit TR. Mammalian telomerase is recruited to telomeres by the TEL patch and NOB regions of shelterin component TPP1. Recent cryo-EM structures of human telomerase reveal that a composite TERT TEN–(IFD-TRAP) domain interacts with TPP1. Here, we generate TERT mutants to demonstrate that a three-way TEN–(IFD-TRAP)–TPP1 interaction is critical for telomerase recruitment to telomeres and processive telomere repeat addition. Single mutations of IFD-TRAP at its interface with TR or the DNA primer impair telomerase catalysis. We further reveal the importance of TERT motif 3N and TEN domain loop $^{99}FGF^{101}$ in telomerase action. Finally, we demonstrate that TPP1 TEL patch loop residue F172, which undergoes a structural rearrangement to bind telomerase, contributes to the human–mouse species specificity of the telomerase–TPP1 interaction. Our study provides insights into the multiple functions of TERT IFD-TRAP, reveals novel TERT and TPP1 elements critical for function, and helps explain how TPP1 binding licenses robust telomerase action at natural chromosome ends.**

## Introduction

Telomeres are nucleoprotein complexes that protect the ends of linear eukaryotic chromosomes from genome instability. Telomeric DNA is bound to the six-protein shelterin complex that protects these ends from being misrecognized as DNA damage sites (Palm & de Lange, 2008). With every cell division, chromosomes undergo progressive shortening because of the directionality of DNA polymerase and the action of exonucleases, thus limiting cellular life span. However, in most proliferative cells like stem cells, these ends are extended with new repeats (GGTTAG in vertebrates) by the reverse transcriptase, telomerase (Greider & Blackburn, 1985, 1989). Most somatic cells do not express telomerase, but an overwhelming majority of cancers depend on the activity of telomerase to divide indefinitely (Kim et al, 1994; Zhang et al, 1999). On the other hand,

several germline mutations associated with telomere shortening have been identified in diseases like dyskeratosis congenita, pulmonary fibrosis, aplastic anemia, and myelodysplastic syndrome (Armanios & Blackburn, 2012; Niewisch & Savage, 2019; Grill & Nandakumar, 2021). Considering how either an excess or a deficiency of telomerase can be detrimental, telomerase function is of enormous clinical significance. Understanding the intricate mechanism of telomerase function at telomeres could guide the development of future therapeutics.

Telomerase is an RNP complex that consists of the catalytic protein subunit TERT (Lingner et al, 1997), which adds new GGTTAG repeats using an internal template residing on the RNA subunit TR (Greider & Blackburn, 1989). TERT consists of four domains – telomerase essential N-terminal domain (TEN), RNA binding domain (RBD), reverse transcriptase (RT) domain (fingers and palm), and the C-terminal extension (CTE) domain (thumb) (Smith et al, 2020). Whereas the catalytic "core" (consisting of RBD, most of the RT, and the CTE) of TERT is similar to those of other RTs, the TEN domain and the TRAP motif within the insertion in fingers domain (IFD) in the RT domain are TERT-specific motifs (Fig 1A and B). The IFD consists of two bracing helices, IFDa and IFDc, that are connected to each other by the TRAP motif (also called IFDb), which consists largely of an extended beta sheet (Wang et al, 2020) (Fig 1A and B). Telomerase has the unique ability to realign with the single-stranded DNA primer to add multiple hexad repeats. During catalysis, the 3' end of the DNA primer forms a duplex with the templating region of TR in the active site of the telomerase. Telomerase performs two types of translocation movements: type I, also called nucleotide addition processivity (NAP), and type II, also called repeat addition processivity (RAP). In type I, the entire primer-template duplex realigns to facilitate the addition of nucleotides to the primer, reminiscent of the mechanism shared by DNA and RNA polymerases and RTs. In type II, which is unique to telomerase, the DNA primer partially dissociates from and realigns with the RNA template to add multiple telomeric-hexad repeats to the primer (Lue et al, 2003; Parks & Stone, 2014). The TEN domain and IFD are two elements in TERT implicated in RAP. The TEN domain is not required for telomerase to synthesize a single repeat (i.e., NAP) but is essential for RAP (Robart & Collins, 2011; Tesmer et al, 2019). IFD was identified as a RAP-regulatory domain in *Saccharomyces cerevisiae* TERT and

Department of Molecular, Cellular and Developmental Biology, University of Michigan, Ann Arbor, MI, USA

Correspondence: jknanda@umich.edu

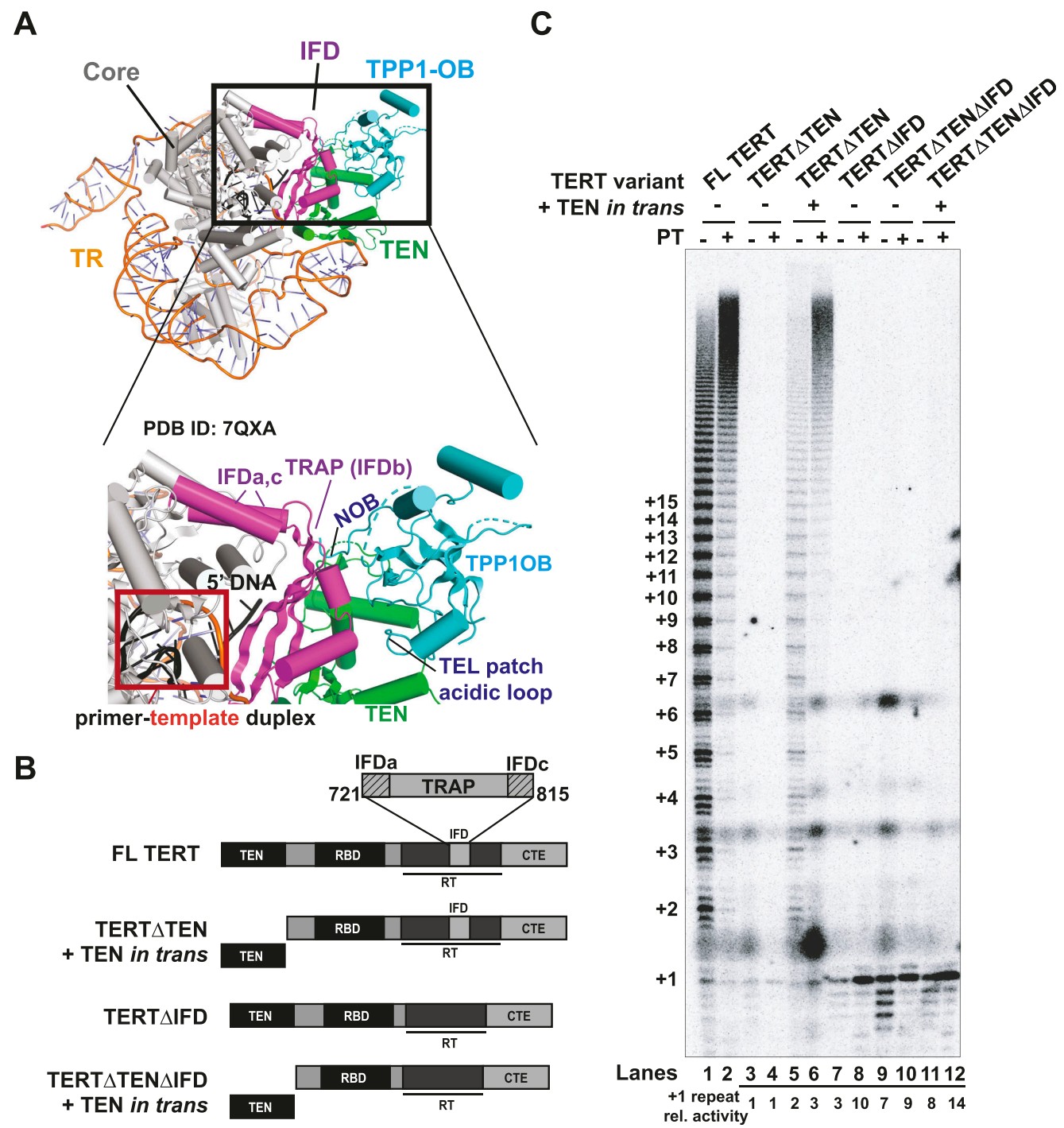

**Figure 1. *Trans* complementation of telomerase activity and processivity with TERTΔTEN and TEN requires the IFD-TRAP.**
**(A)** Overview of the human telomerase cryo-EM structure with hTPP1 (PDB ID: 7QXA). The TERT core (domains excluding TEN and IFD) is shown in grey and TR is shown in orange. A close-up view shows how the IFD-TRAP (magenta) forms a three-way interface with the TEN domain (green) and the NOB and TEL patch regions of hTPP1 OB (Cyan). The red box shows how IFD-TRAP "traps" the primer-template duplex in the active site and potentially interacts with the template and the 5′ end of the primer. **(B)** Domain diagram of human TERT and the deletion constructs used in this study. Domains of TERT, namely, TEN, RBD, RT (which contains the IFD within it), and CTE. The IFD consists of helices IFDa and IFDc that flank the TRAP region (IFDb). Deletion constructs contain a deletion in the TEN, TRAP regions of the IFD-TRAP, or both. **(C)** Direct primer extension assay of TERT variants using 1 μM of a5 primer, and 500 nM of POT1–TPP1 (PT, if added) or protein dilution buffer control (n = 3). Number of hexad repeats added to the primer is indicated on the left of the gel. Reactions were carried out for 1 h at 30°C. Relative intensity (activity) of single repeats (+1) relative to lane 3 is shown below the gel.
Source data are available for this figure.

recent studies have revealed that it plays a similar role in mammalian telomerase (Lue et al, 2003; Chu et al, 2016).

TPP1, a shelterin protein critical for telomerase function, exists as a heterodimer with the telomeric single-stranded DNA-binding shelterin protein POT1 (Baumann & Cech, 2001; Houghtaling et al, 2004; Liu et al, 2004; Ye et al, 2004; Hockemeyer et al, 2006; Wu et al, 2006; Wang et al, 2007). Human TPP1 (hTPP1) binds TERT to recruit telomerase to telomeres (Xin et al, 2007; Abreu et al, 2010; Nandakumar et al, 2012; Grill et al, 2018). In addition, human POT1–TPP1 (referred to as POT1–TPP1 hereafter) binding to TERT stimulates telomerase RAP by improving type II translocation efficiency and reducing the rate of primer dissociation from the template (Wang et al, 2007; Latrick & Cech, 2010). Mutational studies in combination with the recent cryo-EM structures of human TPP1 (hTPP1)-bound human telomerase and *Tetrahymena thermophila* (*Tt*) telomerase holoenzyme provide insights into the three-way protein interface made by TEN and IFD-TRAP domains of telomerase with TPP1 (Jiang et al, 2018; He et al, 2021; Liu et al, 2022; Sekne et al, 2022). Specifically, the TPP1 glutamate (E) and leucine (L)-rich patch (TEL patch) and the N terminus of the OB (oligonucleotide/oligosaccharide-binding) domain (NOB) of TPP1 bind the composite domain formed by the TEN and IFD-TRAP regions of TERT to facilitate telomerase recruitment and TPP1-mediated stimulation of telomerase processivity (Fig 1A) (Nandakumar et al, 2012; Nandakumar & Cech, 2013; Kocak et al, 2014; Dalby et al, 2015; Bisht et al, 2016; Grill et al, 2018; Tesmer et al, 2019).

In addition to revealing how telomerase interacts with TPP1, the recently published 3.2 Å cryo-EM structure of the complex of human telomerase and hTPP1 suggests interactions of the IFD-TRAP and TEN domain with the DNA primer (Fig 1A). The presence of a DNA-binding site on TERT could act as an "anchor site" to aid in RAP (Patrick et al, 2020; Sekne et al, 2022). Whereas previous mutational studies have demonstrated the importance of the C-terminus of the TEN domain (Q169 and residues 170–175) in anchoring DNA, the recent cryo-EM structure shows that the TEN and the IFD-TRAP both interact with the 5′ end of the primer (Jacobs et al, 2006; Wyatt et al, 2007, 2009; Jurczyluk et al, 2011; Sekne et al, 2022). The TEN domain residues Y176/Q177 in the PLYQ motif, along with the IFD-TRAP residue Q794, were shown to be important for activity and RAP in vitro (Sekne et al, 2022). The TRAP is also poised to make contacts with the templating region (K757/F759), near the primer-template duplex that are essential for RAP in vitro (Fig 1A) (Liu et al, 2022; Sekne et al, 2022).

The TEN–(IFD-TRAP) and the fingers domain are structurally related by a pseudo-twofold axis of symmetry and located on opposite ends of the primer–template duplex, prompting the renaming of these regions to fingers-A (previously fingers) and fingers-B (TEN–(IFD-TRAP)) (Wan et al, 2021). In this model, the fingers-A domain facilitates the binding of an incoming dNTP in the active site, whereas the TEN–(IFD-TRAP) stabilizes the primer-template duplex for nucleotide addition and NAP. The IFD-TRAP is also implicated in direct binding to the 3′ end of the templating region of TR. This could help with the ratcheting action required for RAP, as proposed for *Tt* telomerase, but this has not been functionally validated for human telomerase (Jiang et al, 2018).

Here, we systematically evaluate the functional importance of the multiple interfaces of human TERT IFD-TRAP using direct primer extension assays in vitro and telomerase recruitment analysis in

cultured cells. Interpreting our functional data in the context of existing telomerase structural information, we propose that the IFD-TRAP is at the intersection of several functional interactions. Telomerase recruitment to telomeres is IFD-TRAP-dependent, highlighting the importance of the tripartite TEN–(IFD-TRAP)–TPP1 interface. Once at telomeres, the TRAP region of the IFD is poised to make critical contacts with both the 3′ templating region of TR and the 5′ end of the DNA primer, while also coordinating with the TEN to facilitate POT1–TPP1-facilitated primer translocation and RAP (Fig 1A). We also uncover the importance of two IFD-TRAP-adjacent TERT regions, motif 3N in the RT domain and F99-F101 in the TEN domain. Lastly, we show the importance of TPP1 TEL patch amino acid F172, which structurally reconfigures from its buried position in the TPP1 apo OB domain to contact TERT. Our studies reveal a critical hub of functionally important protein–protein and protein–nucleic acid interactions within telomerase and between telomerase and telomeres.

# Results

### *Trans* complementation of telomerase activity and processivity requires the IFD-TRAP

Our previous biochemical studies suggested that the three-way interaction between *Tt* TERT TEN, TERT IFD-TRAP, and p50 (ortholog of TPP1) observed in the high-resolution structures of *Tt* telomerase (Jiang et al, 2015, 2018; He et al, 2021) is conserved in human telomerase (Tesmer et al, 2019; Wang et al, 2020). This observation has recently been confirmed with cryo-EM structures of human telomerase bound to hTPP1 (Liu et al, 2022; Sekne et al, 2022). The telomerase structures provide potential explanations for previous observations, including the reconstitution of human telomerase activity by complementing the TERT core (TERTΔTEN) with the TEN domain in *trans* (Robart & Collins, 2011; Eckert & Collins, 2012) and the stabilization of the TEN domain by hTERTΔTEN (Wu & Collins, 2014). Specifically, the structures suggest that it is the presence of IFD-TRAP within TERTΔTEN that mediates its interaction with TEN to afford *trans* complementation. To test the importance of the IFD-TRAP for mediating telomerase *trans* complementation activity, we engineered IFD-TRAP deletion constructs (Fig 1B) in the context of the *trans* complementation system and subjected them to direct primer extension analysis as reported previously (Tesmer et al, 2019). Telomerase reconstituted from full-length human TERT and TR exhibited characteristic activity and RAP that were further stimulated in the presence of POT1–TPP1, as expected (Fig 1C; lanes 1 and 2) (Wang et al, 2007). TERTΔTEN adds a single repeat in the absence and presence of POT1–TPP1 (+1, lanes 3 and 4) but displayed processive addition of telomeric repeats when the TEN domain was coexpressed in *trans*, as shown previously (lane 5) (Robart & Collins, 2011; Tesmer et al, 2019). Improved RAP was observed with POT1–TPP1 in the context of *trans* complementation, similar to full-length TERT (Fig 1C; compare lanes 5 and 6 to lanes 1 and 2). As observed previously, the activity of telomerase with *trans* complemented TERT does not reach full-length TERT levels even in the presence of POT1–TPP1, suggesting that either TEN or TERTΔTEN level limits the amount of active holoenzyme (Robart & Collins, 2011;

Tesmer et al, 2019). TERTΔIFD, which includes all domains of TERT including TEN, but lacks the IFD-TRAP (aa 735–799) (Fig 1B), was severely compromised for telomerase RAP. Like TERTΔTEN, TERTΔIFD added only a single repeat (Fig 1C; compare lane 7 to lane 3). Despite the presence of the TEN domain, TERTΔIFD was not responsive to the addition of POT1–TPP1 (Fig 1C; compare lanes 7 and 8). The TEN domain was incapable of rescuing RAP of TERTΔTENΔIFD when provided in *trans* even in the presence of POT1–TPP1 (Fig 1C; compare lanes 9 and 10–11 and 12), consistent with IFD-TRAP mediating TEN-TERTΔTEN *trans* complementation. Interestingly, the signal for the single-repeat product was increased 3–14-fold in all reactions lacking the IFD-TRAP (Fig 1C; lanes 7–12) relative to TERTΔTEN (lane 3), consistent with an essential role of IFD-TRAP in type II, but not type I, translocation. The trend for *trans* complementation is not a result of variable TERT protein stability as the levels of the various constructs in the lysates used for primer extension were comparable (Fig S1A). These results are the first to demonstrate that the presence of the IFD-TRAP is required for reconstituting telomerase activity and RAP in *trans*.

### *Trans*-complemented telomerase is recruited to telomeres in a TEN- and IFD-TRAP-dependent manner

Our biochemical analyses reinforce the central importance of the IFD-TRAP in mediating telomerase processivity in vitro. We next evaluated the implications of these results for the recruitment of telomerase to telomeres in HeLa cells. For this, we developed a TERT *trans* complementation-based assay for telomerase recruitment to telomeres, adapting an immunofluorescence-fluorescence in situ hybridization (IF-FISH) method used previously with full-length TERT (Nandakumar et al, 2012; Kocak et al, 2014; Bisht et al, 2016; Grill et al, 2018; Tesmer et al, 2019). Colocalization of the FLAG TERT IF signal and the TR FISH signal was used as a proxy for telomerase RNP assembly, and colocalization of TERT and TR with the telomere PNA FISH signal was used to score telomerase recruitment to telomeres. Although ~95% of telomerase foci were observed to be at telomeres for full length telomerase, a very small percentage of telomerase foci localized to telomeres in the absence of TEN, IFD-TRAP or both (Fig 2A and B). When TEN was added in *trans* to TERTΔTEN, we observed a significant rescue (~35%) in recruitment to telomeres, providing strong evidence for *trans* complementation of telomerase recruitment to telomeres in living cells. Overexpression of hTPP1 in this setup did not further increase the rescue (Fig S1B–D), consistent with the failure of TPP1 to fully rescue telomerase activity when the TEN domain was added in *trans* (Fig 1C). Also consistent with our primer extension experiment, we observed that TERTΔIFD was not recruited to telomeres despite the presence of the TEN domain in this construct (Fig 2A and B). Furthermore, telomerase recruitment to telomeres failed when TERTΔTENΔIFD was *trans* complemented with TEN. Notably, deletion

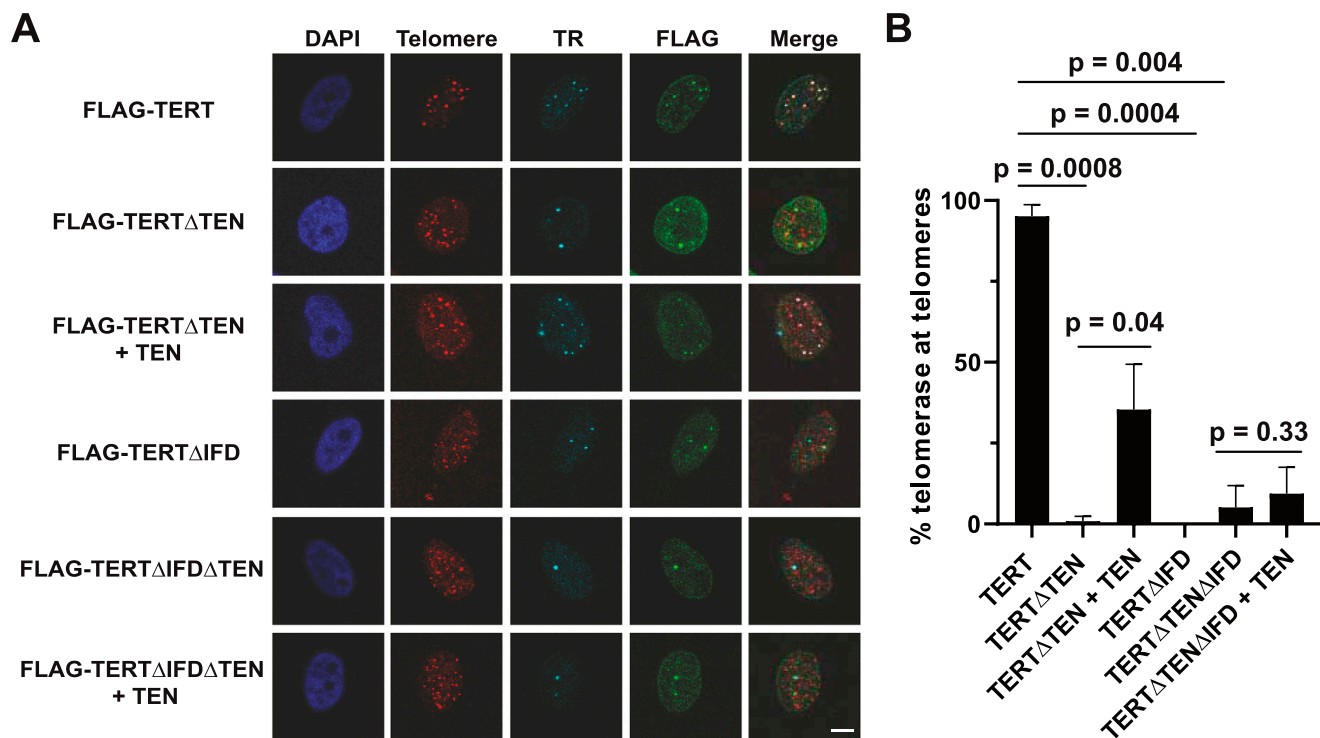

**Figure 2.  *Trans* complementation of telomerase recruitment to telomeres with TERTΔTEN and TEN requires the IFD-TRAP.**
**(A)** Representative images of HeLa cells showing FLAG IF foci representative of TERT variants. FISH was used to detect telomeric DNA and TR. DAPI was used to stain nuclei. Merge shows the colocalization of TERT and TR with telomeres (scale bar = 5 μm). **(B)** Quantification of percentage of assembled telomerase (colocalized TERT and TR foci) at telomeres is shown for each transfection condition (n = 3). Student's Two-tailed *t* test was used to calculate *P*-values. At least 80 assembled telomerase foci for each condition were counted.
Source data are available for this figure.

of the TEN or IFD-TRAP did not affect the characteristic colocalization of TERT and TR in Cajal bodies (Venteicher et al, 2009), implying that this interface is dispensable for telomerase RNP assembly (Fig 2A and B). Indeed, neither of these domains are required for telomerase to add a single repeat in primer extension analysis (Fig 1C). Given that the interaction between TPP1 and TERT defines telomerase recruitment to telomeres (Abreu et al, 2010; Zhong et al, 2012), our data demonstrate the interaction between TEN, IFD-TRAP, and TPP1 that drives telomerase recruitment to telomeres can be achieved with each of these elements presented in *trans*, highlighting the robustness of this three-way interaction in the cell.

## Mutagenesis screen to evaluate the importance of IFD-TRAP interfaces

The DNA-bound *Tt* telomerase holoenzyme structures revealed that the TRAP region of the IFD is poised to interact with not just the TEN domain and the *Tt* equivalent of TPP1 OB domain, p50-OB, but also regions of TR and the DNA primer. The homology model of human telomerase bound to hTPP1 generated using a *Tt* telomerase structural template (as reported previously; [Tesmer et al, 2019]) showed that the human IFD-TRAP could similarly "trap" the primer-template duplex in the active site. Consistent with this notion, it has also been shown that TERTΔTEN can protect the 5′ end of a single-stranded DNA primer on its own (Wu & Collins, 2014). Based on these structural and biochemical findings, we wondered whether human IFD-TRAP may participate in direct interactions with TR and/or DNA important for telomerase activity. As the current study started before the publication of the recent cryo-EM structures of human telomerase, we used our homology model (Tesmer et al, 2019) to identify potential nucleic acid-binding IFD-TRAP residues and engineered mutations at nine conserved residues in this region. We introduced these mutations in the context of the *trans* complementation system to potentially separate phenotypes arising from defective TEN-TRAP-nucleic acid interaction versus TERT–TPP1 interaction. Basic (lysine [K], arginine [R], and histidine [H]) and hydrophobic (phenylalanine [F] and leucine [L]) residues were mutated to glutamate (E) to disrupt putative interactions with nucleic acid, whereas the negatively charged D788, suspected to bind the TEN domain, was mutated to the oppositely charged arginine (R) residue. We confirmed the expression of WT and mutant FLAG-TERTΔTEN and Myc-TEN in HEK 293T cell extracts before the primer extension assay (Fig S2A).

All the mutants had striking effects on activity and processivity, with the exception of H762E, which showed 86% processivity with POT1–TPP1 and 68% activity relative to WT(Fig 3A, quantified in Fig 3B and C). The cryo-EM structure revealed that H762 is a polar, surface residue of the IFD-TRAP that is not proximal to any of its binding partners, explaining the mild phenotype for this mutant (Fig S3C). In contrast, F776E and L786E showed greatly reduced telomerase activity (Fig 3A and C) despite being expressed at levels comparable to WT TERT (Fig S2A). This observation is consistent with F776 and L786 representing key hydrophobic residues that form the core of the IFD-TRAP domain (Fig S3C), highlighting their importance in determining the proper folding and function of this region. D788R showed very weak telomerase activity compared with WT (24% of

WT; Fig 3A and C), but close to WT-like POT1–TPP1-stimulated processivity (64%; Fig 3B). D788 is located adjacent to the N-terminal helix of the TEN domain (Fig S3C), suggesting that the defect in *trans* complementation could arise from a weakened TEN–(IFD-TRAP) interface. We were unable to establish statistical significance for potential differences in activity because of the low signal-to-noise ratio associated with the *trans* complementation system.

## Template- and primer-adjacent residues of the IFD-TRAP are critical for telomerase *trans* complementation

The human telomerase cryo-EM structures reveal the IFD-TRAP as a structural clamp that encloses the primer-template duplex in the active site of the enzyme (Fig 1A). However, it is possible that in addition to physically "trapping" the primer-template duplex to promote telomerase processivity, the IFD-TRAP regions may also make side chain-specific contacts to the template and primer to assist the active site with catalysis. In agreement with this idea, single mutants K749E, R756E, K757E, F759E, and K760E all showed severe activity and POT1–TPP1-stimulated processivity defects in the *trans* complementation system (Fig 3A–C). To rule out that the activity phenotype arose from misfolding of TERT (or the IFD-TRAP portion of TERT) and/or misassembly of the telomerase RNP, we transiently transfected full length TERT containing the mutations K749E, R756E, and F759E, along with TR, into HeLa cells and quantified telomerase recruitment to telomeres using IF-FISH. Telomerase containing these IFD-TRAP mutations was efficiently recruited to telomeres at levels comparable to WT (Fig S2B and C), strongly suggesting that the telomerase activity phenotypes result from defects in catalysis. Inspection of these residues within the structure of human telomerase revealed that residues K749 and R756 in the IFD-TRAP bind the 3′ end of the TR template at residues C56 and A55, respectively (Figs 3F and H and S3B). Specifically, each of these positively charged sidechains forms a pi–cation stacking interaction with its contact RNA base. In contrast, K760 is not close to TR or the primer (Fig S3C), suggesting that its contribution to activity and processivity may be indirect or potentially arise from a transient nucleic acid interaction occurring at a stage of catalysis (e.g., translocation) not captured in the cryo-EM structure.

The study reporting the human telomerase structure bound to a DNA primer described K757 and F759 as being part of the anchor site of telomerase on DNA and demonstrated that double mutant K757A/F759A, like our single mutants K757E and F759E, was defective in telomerase activity and processivity (Sekne et al, 2022). Both these residues are positioned close to the DNA sequence immediately upstream (i.e., 5′) of the primer-template duplex, with K757 poised to form an ionic interaction with the phosphodiester backbone of the DNA and F759 potentially stacking with the base/s of the single-stranded DNA (Figs 3F–H and S3A and B). The DNA contacts made by the IFD-TRAP (K757, F759, and Q794) and TEN domain ($^{174}$PLYQ$^{177}$) are likely part of the anchor site required for telomerase RAP (Fig 3F–H).

Close to the contacts made by the TEN and TRAP with the 5′ end of the DNA primer, there is a highly basic loop in the RT domain of human TERT ($^{636}$MDYVVGARTFRREKRAER$^{653}$) called motif 3N in the context of *Tt* TERT (Fig 3G; [He et al, 2021]). To evaluate if motif 3N is involved in processivity regulation, we tested if its deletion affects

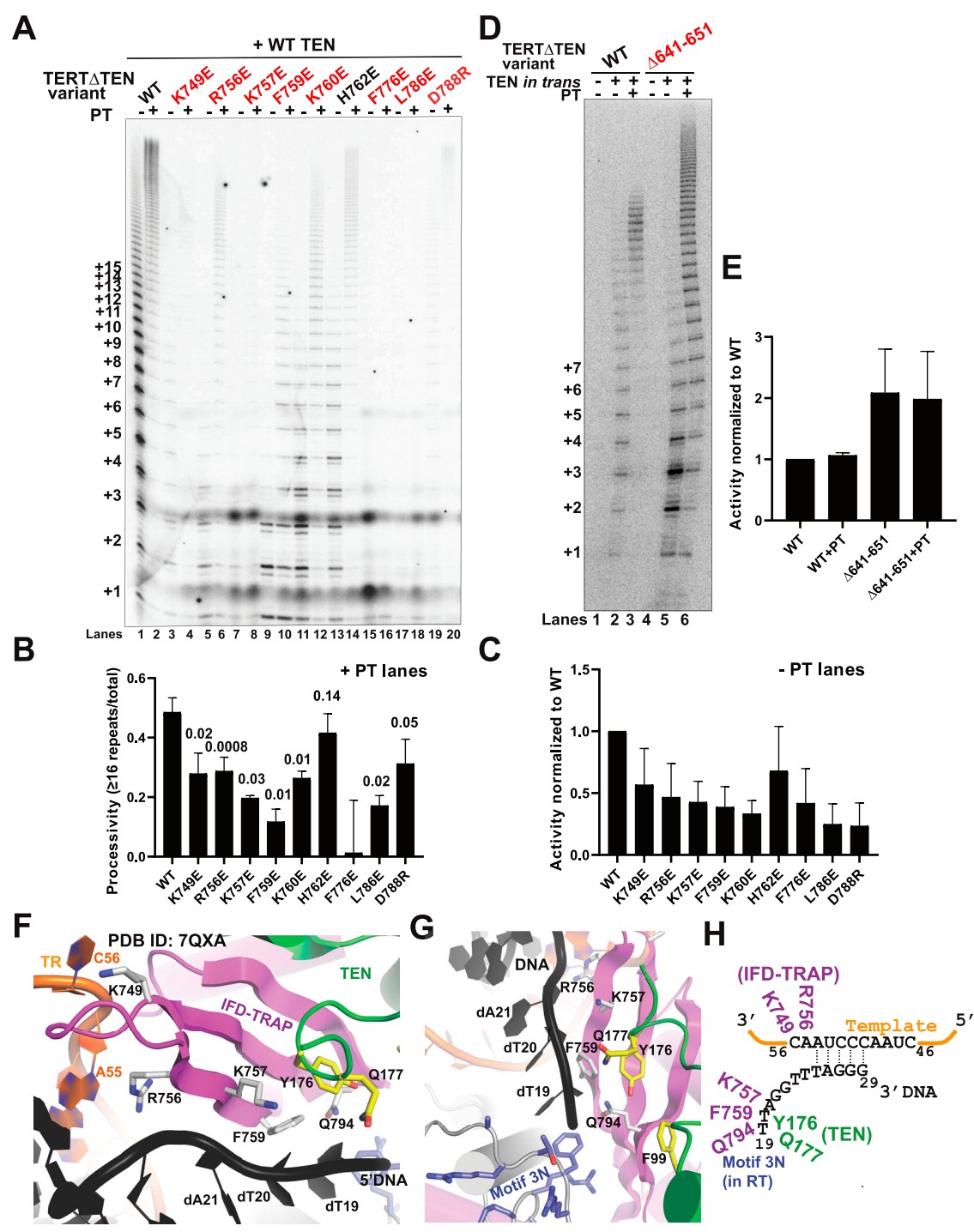

**Figure 3. Mutations in the IFD-TRAP and adjacent Motif 3N affect TEN-TERTΔTEN *trans* complementation.**
**(A)** Representative gel showing direct primer extension assay products of extracts containing WT TEN with TERTΔTEN variants (WT and IFD-TRAP mutants) and TR. Reactions were incubated at 30°C for 1 h with 500 nM of POT1–TPP1 (PT) in the even-numbered lanes, or protein dilution buffer in the odd-numbered lanes. **(A, B, C)** Quantification of the percent processivity (B) and total activity (C) from three independent experiments of which panel (A) is representative. Total activity for the mutants was calculated by quantifying the intensity of the entire lane and dividing it by the intensity of the lane containing WT telomerase. Processivity was calculated by dividing the cumulative intensity of bands representing 16 repeats and greater by the total activity. **(A, B)** P-values in panel (B) were calculated using *t* test for comparisons against WT + PT (lane 2 of panel (A)) and are indicated above each bar graph. **(D)** Primer extension assay products of WT or Δ641–651 TERTΔTEN + TR in the presence or absence of TEN and/or POT1–TPP1 (PT). **(E)** Total activity (normalized to WT) of data from three independent experiments (of which D is representative). **(F)** Close-up view of cryo-EM structure of human telomerase bound to telomeric DNA and hTPP1 (PDB ID: 7QXA), showing IFD-TRAP residues K749 and R756 contacting TR residues C56 and A55, respectively (residues are in grey and IFD-TRAP is in magenta). The 5' end of the DNA (black) is trapped in place by the IFD-TRAP. **(F, G)** Orthogonal view of panel (F) showing IFD-TRAP (K757, F759, Q794), TEN domain ($^{174}$PLYQ$^{177}$ motif; residues are highlighted in yellow and TEN is in green), and motif 3N (aa 741–751; residues are

*trans* complementation. We also tested it in the context of POT1–TPP1 as the 5′ end of the DNA leads into the DNA-binding domain of POT1 in the cryo-EM structure reconstruction (Sekne et al, 2022). We expressed WT and Δ641–651 variants of FLAG-TERTΔTEN along with Myc-TEN in HEK 293T cells (Fig S2D) and performed *trans* complementation-based primer extension with the cell extracts. We observed that *trans* complementation of TERTΔTENΔ641–651 with TEN resulted in a lower RAP compared with WT (Fig 3D; compare lanes 2 and 5; quantified in Fig S2E), Surprisingly, the *trans*-complemented mutant was more active than WT telomerase (by ~twofold, Fig 3E). The contrasting trends for the mutant's activity and processivity result from an increased accumulation of shorter extension products, which persists even upon stimulation of processivity by POT1–TPP1 (Fig 3D; see quantification of processivity and fraction of short-extension products in Fig S2E and F). Introducing this deletion in the FLAG TERT (full length) backbone resulted in efficient colocalization with TR and telomeres, supporting the integrity of the telomerase recruitment interface in the presence of this mutation (Fig S2G and H). Our results demonstrate the importance of (IFD-TRAP)–nucleic acid interactions near the active site of telomerase and an autoinhibitory role of the TEN–(IFD-TRAP)-adjacent motif 3N in telomerase-mediated chromosome end extension.

## TEN domain $^{99}$FGF$^{101}$ loop is important for telomerase recruitment to telomeres and POT1–TPP1-mediated processivity stimulation

The cryo-EM structure of human telomerase with POT1–TPP1 and DNA (PDB ID: 7QXS) and past functional studies have demonstrated that the TERT TEN–(IFD-TRAP) binds TPP1. Intriguingly, TEN domain residue G100, which was previously shown to be critical for POT1–TPP1-mediated processivity stimulation of human telomerase (Zaug et al, 2010), is not close to TPP1 in the solved structures (Liu et al, 2022; Sekne et al, 2022). How a glycine residue distal from the TPP1-binding surface influences the functional interaction with POT1–TPP1 is unknown. Flanking G100 are two conserved aromatic residues F99 and F101 (Fig S3D), which are surrounded by anchor site residues in the TEN–(IFD-TRAP), the 5′ end of the primer, motif 3N, and the DNA-binding domain (DBD) of POT1, raising the possibility that they may be involved in nucleic acid and/or POT1 binding (Fig 4C). To gain more insights into their contribution/s to telomerase function, we introduced F99V and F101V into the FLAG-TERT (full-length) backbone, transiently transfecting them into HEK 293T cells alongside WT FLAG-TERT and performed telomerase-mediated primer extension analysis (Fig 4A and B). The expression of F99V and F101V TERT is similar to that of WT TERT (Fig S3E). WT TERT + TR shows the characteristic increase in RAP in the presence of POT1–TPP1 (or a fusion of hTPP1 OB and human POT1 DBD called OB-DBD, which can be used as a substitute for POT1–TPP1 [Grill et al, 2018]) that is statistically significant (Fig 4A; compare lane 1 with lanes 2 and 3; quantified in Fig 4B). However, F99V is defective in POT1–TPP1 (and OB-DBD)-mediated processivity stimulation as

indicated by the lack of a statistically significant increase in processivity in the presence of the shelterin subcomplex (Fig 4A, compare lane 7 with lanes 8 and 9; quantified in Fig 4B). F101V exhibited an intermediate phenotype (Fig 4A and B).

As we observed a defect in POT1–TPP1-mediated processivity but not in the basal activity or processivity of telomerase, we wondered if F99 and F101 in the TEN domain are essential for telomerase recruitment to telomeres. In stark contrast to WT telomerase, the coexpression of FLAG-TERT F99V or F101V with TR either abrogated (F99V; Fig 4D and E) or greatly diminished (F101V; Fig 4F and G) colocalization of FLAG foci with TR foci at telomeres, with no effect on telomerase assembly. These results separate the contributions of the TEN domain residue F99 from those of the primer-template adjacent IFD-TRAP residues K749, R756, F759, K760, and motif 3N, which were important for telomerase activity but not telomerase recruitment to telomeres. The positioning of F99 (or G100) at the critical intersection of TEN-TRAP-DNA and POT1-DBD in the cryo-EM structure (PDB ID: 7QXS, Fig 4C) supports the possibility that POT1, in the context of POT1–TPP1, plays a direct role in telomerase recruitment to telomeres. Together, our studies suggest that the POT1–TPP1 interaction interface with telomerase may be more elaborate than previously suggested by mutational studies (Nandakumar et al, 2012; Grill et al, 2018; Tesmer et al, 2019).

## hTPP1 TEL patch residue F172 rearranges structurally to bind telomerase and is important for the mouse-human species specificity of the telomerase-TPP1 interaction

The recent cryo-EM structures elegantly explain the plethora of previous mutagenesis data that led to the identification of the TEL patch and NOB of TPP1 as major determinants of telomerase recruitment to telomeres (Zaug et al, 2010; Nandakumar et al, 2012; Kocak et al, 2014; Schmidt et al, 2014, 2016; Bisht et al, 2016; Grill et al, 2018; Liu et al, 2022; Sekne et al, 2022). Based on the inspection of the cryo-EM structure of hTPP1-bound human telomerase, we find that the strictly conserved TEL patch acidic glutamate residues E168, E169, and E171 on hTPP1 play a critical role in bolstering the TEN–(IFD-TRAP) interface. Specifically, the negative charge of each glutamate sidechain caps the N-terminus of a particular TERT helix, stabilizing the partial positive charge from the helical dipole (Fig 5A). Of the three converging TERT helices, two are from the TEN domain (aa 136–143 and aa 45–53) and one is from the IFD-TRAP (aa 772–784). Interestingly, the binding of hTPP1 to human TERT is accompanied by a conformational change in the apo structure of hTPP1 OB in the region immediately after these acidic TEL patch residues that causes the previously buried hydrophobic residue F172 to become sandwiched between two helices in the TEN domain (PDB ID: 2I46; compare Fig 5C and D [Liu et al, 2022; Sekne et al, 2022]).

Despite appearing to play a critical role in this structure, F172 is not conserved in mouse TPP1 (corresponding residue is a leucine, L84), but is conserved in other mammalian homologs. We have previously exploited the fact that RAP of human telomerase is not

highlighted in lavender) adjacent to dA21, dT20, and dT19 of the DNA primer (PDB ID: 7QXA). **(H)** Schematic of primer-template duplex showing the potential interactions made by IFD-TRAP, TEN, and Motif 3N with the template and/or DNA.
Source data are available for this figure.

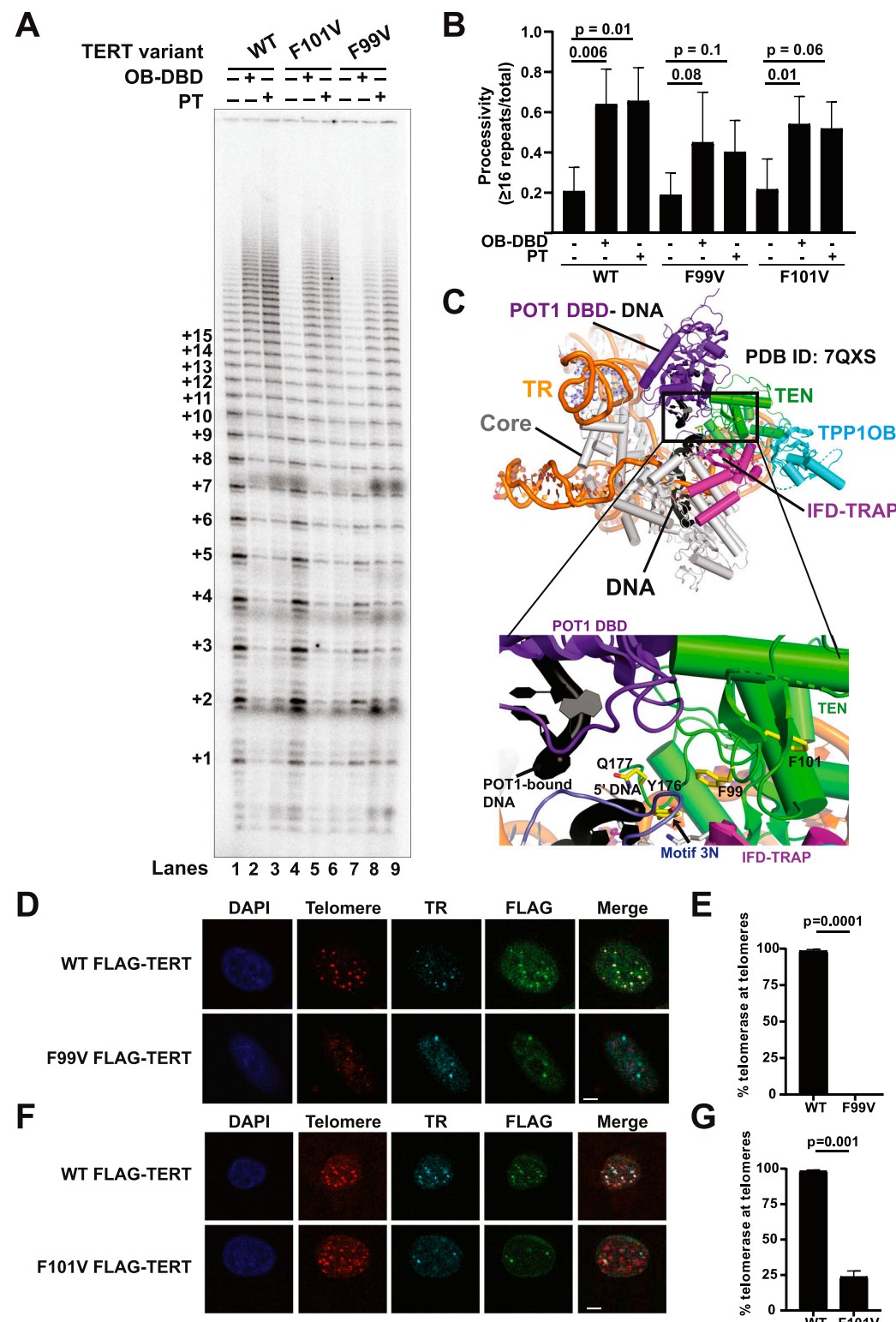

**Figure 4. TEN domain ⁹⁹FGF¹⁰¹ loop is essential for POT1–TPP1 processivity stimulation and telomere recruitment of telomerase.**

**(A)** Representative gel showing primer extension products of WT, F101V, and F99V TERT, and TR with 1 µM primer a5 for 20 min at 30°C. Reactions represented in lanes 2, 5, and 8 contained 500 nM OB-DBD; lanes 3, 6, and 9 contained 500 nM PT; and lanes 1, 4, and 7 have protein dilution buffer added as a control. **(A, B)** Processivity from three replicates of which panel (A) is representative, calculated as cumulative intensity of bands representing 16 repeats or greater divided by the total activity in the lane, excluding bands representing repeats 1 and 2. *t* test was used to calculate *P*-values for comparing +OB-DBD and +PT lanes to their protein buffer (−) controls. *P*-value < 0.05 suggests statistically significant stimulation of telomerase processivity by POT1–TPP1. No significant differences were observed in the basal processivities of WT,

stimulated by mouse TPP1 to demonstrate that the NOB is a key feature of the TPP1–telomerase interface, which has recently been structurally validated by its cryo-EM structure (Grill et al, 2018; Liu et al, 2022; Sekne et al, 2022). Although the NOB is conserved in most mammals, it is not conserved in mice; incorporating the human NOB residues into mouse TPP1 allows it to stimulate human telomerase processivity (Grill et al, 2018). We asked whether substituting mouse TPP1 L84 with its equivalent F172 residue in hTPP1 would allow for a similar gain of function phenotype, allowing the mouse TPP1 protein to better stimulate human telomerase processivity.

To focus our studies on the effect of TPP1 OB in these experiments, we used OB-DBD constructs instead of POT1–TPP1. OB-DBD variants in which the human or mouse TPP1 OB domains are connected to the DBD of human POT1 (Fig 5B) (Grill et al, 2018) were recombinantly expressed in *E. coli* and purified (Fig 5E). Primer extension analysis was performed on HEK 293T extracts containing TERT and TR (Fig 5F) and quantified for effects on activity (Fig 5G) and processivity (Fig 5H). Telomerase processivity was stimulated by the addition of either POT1–TPP1 or hOB-DBD, but not mOB-DBD (compare lanes 2 and 3 with lane 4) as reported previously (Tesmer et al, 2019). Also consistent with previous studies, mOB-DBD harboring the human TPP1 NOB sequence in place of the mouse NOB sequence, m(hNOB), regained the ability to stimulate processivity (Fig 5F and H; lane 5). mOB-DBD with an L84F mutation, m(L84F), stimulated a higher level of telomerase processivity (lane 6) compared with mOB-DBD (lane 4) (Fig 5F and H). When both the human NOB and the TEL patch F172 residues were incorporated into equivalent positions in mOB-DBD (m(hNOB, L84F); Fig 5F and H; lane 7), we could establish a statistically significant increase in human telomerase processivity compared with telomerase alone (lane 1) or that supplemented with mOB-DBD (lane 4) (Fig 5H). Although establishing statistical significance was challenging for total activity levels because of the larger variation in these measurements, the activities from reactions containing mOB-DBD (lane 4) were reproducibly lower than those containing hOB-DBD (lane 3), a defect that could be rescued by incorporating both the human NOB and L84F in mOB-DBD (m(hNOB, L84F); Figs 5F and G; lane 7).

The importance of F172 was further evaluated by mutating the hOB-DBD construct to contain the mouse leucine residue in lieu of human F172 (F172L, Fig S4A). The effect of this single mutation was subtle (Fig S4B; compare lanes 1–3), but when mutated to also contain the mouse NOB region, it resulted in a clear suppression of the stimulatory effects of hOB-DBD (Fig S4B; compare lanes 1 and 2 with lane 5; h(mNOB, F172L); quantified in Fig S4C and D). Our results demonstrate that F172 is a previously unappreciated TEL patch residue important for contacting telomerase and contributing to the observed species specificity of human over mouse TPP1 for stimulating human telomerase.

# Discussion

### The IFD-TRAP participates in important protein–nucleic acid interactions during telomerase catalysis

We identified several human IFD-TRAP mutants in the TRAP region, specifically, K749E, R756E, K757E, and F759E, which compromised the ability of telomerase to synthesize telomeric repeats in the context of TEN–TERTΔTEN *trans* complementation. The observed phenotype in conjunction with cryo-EM structures of human telomerase suggests that these basic and aromatic IFD-TRAP residues bind nucleic acid to facilitate catalysis. Our results are consistent with K757 and F759 contributing to the anchor site of TERT as recently suggested (Sekne et al, 2022). However, our results demonstrate that anchoring of telomerase to telomeric DNA is separable from the stable recruitment of telomerase to telomeres, as TERT K757E and F759E localize to telomeres like WT protein. We also identified and validated for the first time TR-binding residues in the human TERT IFD-TRAP, namely K749 and R756. These residues contribute to catalysis but are not important for telomerase RNP formation or telomerase recruitment to telomeres (Fig 3F–H). None of these mutations were severe enough to produce the "single repeat" phenotype observed upon the IFD-TRAP (or TEN) deletion. We reason that the phenotype of the single mutants is more subtle than that of the full deletion because they impact some (e.g., DNA or RNA binding), but not other (e.g., TEN and TPP1 binding) functions of the IFD-TRAP.

### A complex role for human TERT motif 3N in telomerase activity and processivity

The basic motif 3N of human telomerase is proximal to the 5′ end of the primer as it emerges out of the active site (Fig 3G and H). In the context of *trans* complementation with the TEN domain, we found that the TERTΔTENΔ641–651 motif 3N deletion mutant increases telomerase activity, in agreement with a previous study that used a smaller deletion in this region in the context of full length TERT (Δ643–649) (Xie et al, 2009). TERTΔTENΔ641–651 reduced telomerase processivity in the presence and absence of POT1–TPP1. These defects are reminiscent of the compromising effect of single mutations or deletions in TERT motif 3N on *Tt* telomerase activity (He et al, 2021). However, the longest products synthesized by this mutant were longer than those synthesized by WT telomerase under the same conditions. The observation of a larger spread in product length despite lower overall processivity could be explained by the ability of the mutant to add more nucleotides (and hence, repeats) per unit time during processive runs, even though the frequency of such runs is less than that of WT TERT. These data

F101V, and F99V (*P*-values not shown). **(C)** Overview of the human telomerase cryo-EM structure (PDB ID: 7QXS) with a close-up of the region that includes residues F99 and F101 (highlighted in yellow) of the TERT TEN domain (green). F99 and F101 are at the intersection of motif 3N (lavender), IFD (magenta), DBD of POT1 (purple) bound to DNA, and 5′ end of DNA (black). **(D, F)** Representative images of IF-FISH of FLAG-TERT (panel D: WT or F99V; panel F: WT or F101V) and TR transfected in HeLa cells, where FLAG is detected with anti-FLAG antibody, TR and telomeres are detected by FISH, and nuclei are detected by DAPI staining. Merge shows the colocalization of TERT and TR foci at telomeres (scale bar = 5 μm). **(D, E, F, G)** Quantification of telomerase recruitment to telomeres of data for which panels (D, F) are representative (n = 3). Student's Two-tailed *t* test was used to calculate *P*-values. At least 120 assembled telomerase foci for each condition were counted. Source data are available for this figure.

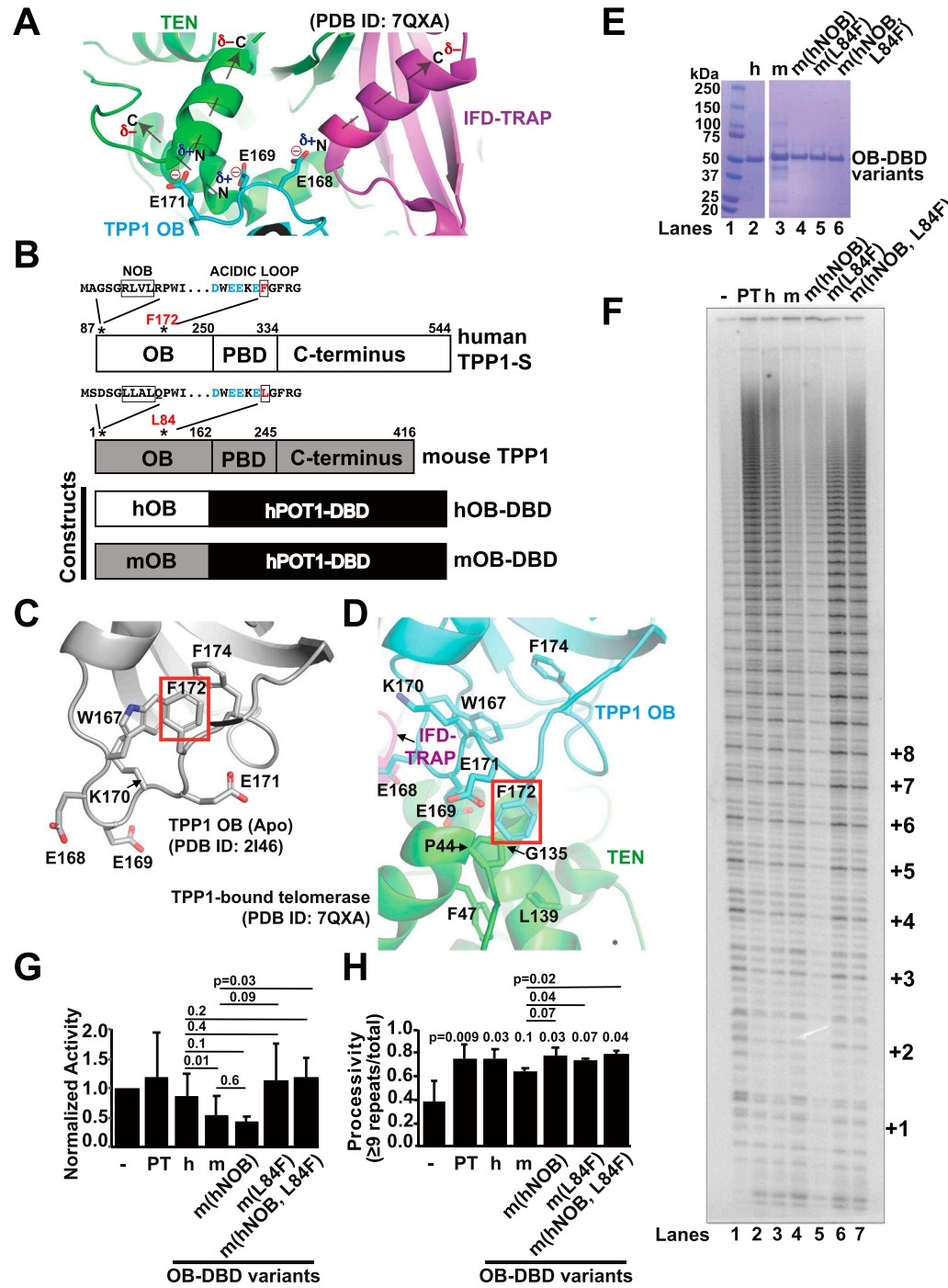

**Figure 5. hTPP1 F172 is a new TEL patch residue that contributes to species specificity of human telomerase for human over mouse TPP1.**
**(A)** Negatively charged sidechains of TEL patch residues E168, E169, and E171 cap the N-termini of two TEN helices and one IFD-TRAP helix, interacting electrostatically with the helical dipoles; PDB ID: 7QXA. The minus sign in red indicates negative charge and plus sign in blue indicates positive charge; $\delta$ indicates partial charge. Dashed arrows indicate the directionality of each TEL-patch-interacting TERT helix and its helical dipole. **(B)** Domain diagrams of human (h) and mouse (m) TPP1 (OB domain, POT1-binding domain, and the C-terminus; amino acid boundaries noted) are shown alongside the two primary OB-DBD constructs that link either the hOB or the mOB to the DNA-binding domain of human POT1 (DBD). Shown above the domain diagrams are TPP1 OB domain sequences of the NOB (boxed) and TEL patch acidic loop (conserved acidic residues are in cyan and human F172/mouse L84 are boxed and in red). **(C, D)** Comparison of the TPP1 acidic loop in the crystal structure of the apo form of hTPP1 OB ((C); PDB ID: 2I46, grey with F172 highlighted in red box) and the cryo-EM structure of hTPP1 (cyan with F172 in red box) bound to telomerase ((D); PDB ID: 7QXA; TERT TEN domain shown in green and IFD shown in magenta). **(E, F)** Coomassie blue-stained SDS–PAGE gel of purified OB-DBD proteins that were subsequently subjected to telomerase primer extension analysis (panel (F)). Proteins correspond to hOB-DBD and mOB-DBD (shown in B) and three variants of mOB-DBD that swap in the human NOB region, human F for mouse L84 in the acidic loop, or both. **(F)** Primer extension analysis of telomerase with protein dilution buffer control (lane 1) or supplemented with POT1–TPP1 (PT, lane 2) or the indicated OB-DBD variants (lanes 3–7). **(F, G, H)** Activity (normalized to extracts alone, lane1) (G) and processivity (H) for three replicates

suggest that motif 3N has contrasting roles in human telomerase catalysis. Although a structural basis for these activities remains to be established, we speculate that the dampening of human telomerase activity by motif 3N may help to optimize the balance between nucleotide addition and steps important for RAP such as template translocation.

### The $^{99}$FGF$^{101}$ loop mediates telomerase recruitment despite functioning at a distance from TPP1

The observed ~7–9 Å density for the POT1 DBD bound to telomeric ss DNA upstream of the TERT–TPP1 interface and the anchor site in the cryo-EM structure (PDB ID: 7QXS) raises the intriguing possibility that the telomerase recruitment interface may extend beyond TPP1 to include POT1 (Sekne et al, 2022). Prior studies on TERT G100V (Zaug et al, 2010) and our new data with TERT F99V lend support to this model. Despite being far away from the TEN–(IFD-TRAP)–TPP1 OB interface (Fig 4C), mutations in this region of the TEN domain reduce the stimulation of telomerase processivity by POT1–TPP1 and cause severe defects in telomerase recruitment to telomeres in cells. Given the proximity of F99 to the DNA-bound POT1 DBD modeled in the human telomerase cryo-EM structure (PDB ID: 7QXS; Fig 4C), it is tempting to speculate that telomerase recruitment to telomeres involves binding of both POT1 and TPP1 to TERT. Intriguingly, recent work on mouse POT1 paralogs revealed that POT1B, but not POT1A, facilitates TPP1-dependent telomerase recruitment to telomeres. However, specific residues in the TPP1-binding domain (not resolved in cryo-EM structures of POT1–TPP1–DNA-bound telomerase), rather than the DBD, of POT1B, were shown to be critical for TPP1-dependent telomerase recruitment to telomeres (Gu et al, 2021). A structure resolving both human telomerase and full-length POT1 and TPP1 could provide deeper insights into the role of POT1 in TPP1-mediated telomerase recruitment and processivity stimulation.

### TPP1 OB stabilizes the composite TERT TEN–(IFD-TRAP) domain

In addition to showcasing a TEN–(IFD-TRAP)–TPP1 OB three-way interface, cryo-EM data have revealed a direct correlation between TEN conformational stability and the presence of bound TPP1 OB (Liu et al, 2022; Sekne et al, 2022). In the cryo-EM data, we also find a structural basis for how the TEL patch and NOB regions of TPP1 reinforce the TEN–(IFD-TRAP) interaction. The NOB forms an extended structure that contacts both the TEN and IFD-TRAP regions, forming an extensive interface with TERT. Another key aspect in the binding of TPP1 to TEN–(IFD-TRAP) involves electrostatic interactions formed by the acidic loop of the TEL patch with the helical dipoles on the TEN and IFD-TRAP (Fig 5A). This represents a unique electrostatic interaction between the sidechain of an amino acid and a secondary structure of its interaction partner, as opposed to the more commonly observed salt-bridge formation between amino acids with oppositely charged sidechains. This charge

stabilization helps TEN and IFD-TRAP to coalesce and also provides a structural explanation for previous mutagenesis studies that show E→D mutations (that retain negative charge), but not isosteric E→Q mutations (neutral in charge) for E168, E169, and E171 can sustain POT1–TPP1-mediated telomerase processivity stimulation (Bisht et al, 2016). One of the TPP1-bound human telomerase cryo-EM studies implicated TEL patch acidic loop glutamate E168 forming a salt-bridge with R774 of TERT (Liu et al, 2022) (PDB ID: 7TRE), but this was not observed in another study (Sekne et al, 2022) (PDB ID: 7QXA). The relevance of this interaction is unclear as R774 is not strongly conserved in mammalian TERT homologs (substituted by glycine in rat and mouse homologs) although the TEL patch glutamate residues are. Thus, only the TERT helical dipole interactions with the TPP1 TEL patch glutamate residues seem strictly conserved across all mammals.

Apo and telomerase-bound TPP1 structures demonstrate that hTPP1-F172, which follows the acidic loop of the TEL patch, undergoes dramatic conformational changes from its buried form in apo hTPP1 to form a central crux of interactions with the TEN domain (Fig 5C and D). The importance of this critical contact is shown through our gain of function studies that exploit the fact that human telomerase is stimulated by human and not mouse TPP1: substitution of the anomalous mouse leucine at this position, L84, with the largely conserved phenylalanine, F172 in human, facilitates mouse TPP1 to stimulate human telomerase. F172 and the NOB region of hTPP1 constitute the major reason why mouse TPP1 cannot stimulate human telomerase (Fig 5F). Together with other studies on the NOB and TEL patch, we have extensive biochemical support for the interactions observed in the cryo-EM structure between TPP1 and the TEN–(IFD-TRAP) region of telomerase.

### Dependence on TPP1 for TEN–(IFD-TRAP) stabilization can ensure telomerase activation only at natural chromosome ends

We propose that the dependence of the conformational stability of the TEN domain on TPP1 binding serves as an additional quality control step that ensures the activation of telomerase specifically at telomeres, curbing telomerase action at non-telomeric substrates such as genomic double-stranded breaks. Ensuring specificity for telomeres becomes especially important as this substrate is highly limiting in mammalian cells (92 chromosome ends in a diploid human cell). Our model predicts that such quality control may be relaxed in cells with a high concentration of telomeres (relative to the size of the genome). Ciliates such as *T. thermophila* contain a plethora of natural chromosome ends (with telomeres) in their macronucleus (Gorovsky, 1980). Indeed, the three-way interface between the TEN–(IFD-TRAP)–TPP1 is conserved as the p50-TEN–(IFD-TRAP) interface in *Tt* telomerase, but p50 is a telomerase holoenzyme subunit rather than a telomeric protein (Min & Collins, 2009). A similar distinction can be made for the two popular yeast models, *S. cerevisiae* (*S. cerevisiae*) and *Schizosaccharomyces pombe* (*S. pombe*), although the difference in telomere number is

---

of the primer extension assay of which panel (F) is representative. *P*-values from Student's two-tailed *t* test comparing telomerase processivity alone with the indicated sample are shown above the bar graph.
Source data are available for this figure.

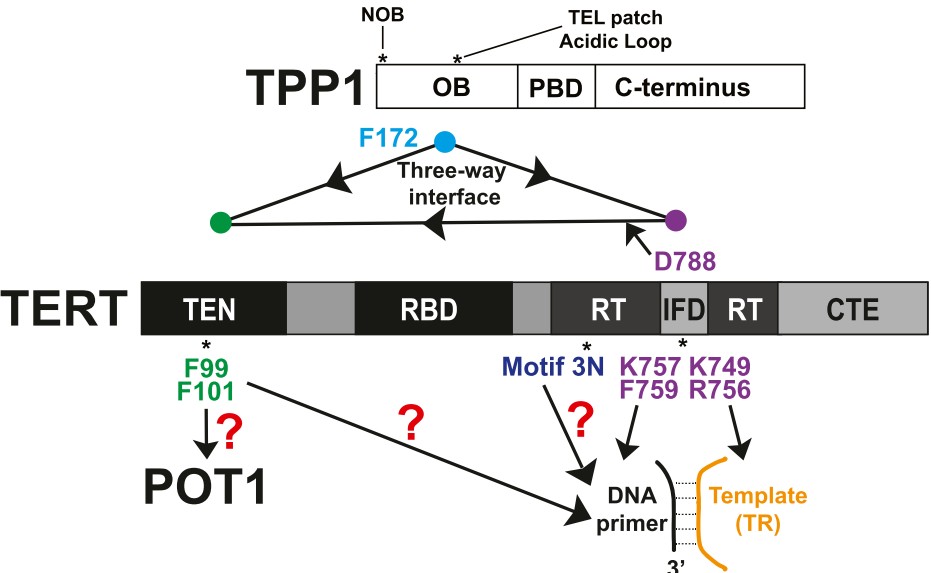

**Figure 6. Schematic of the predicted functions of the important residues analyzed in this study.**
The TEN–(IFD-TRAP)–TPP1 interface is facilitated by the TEL patch and NOB on TPP1. Newly identified TEL patch residue F172 undergoes a conformational change and is poised to interact with the TEN domain. TEN domain residues F99 and F101 potentially interact with POT1 and/or the DNA primer. Motif 3N in the RT is situated close to the IFD-TRAP and the DNA primer and regulates telomerase catalysis. IFD-TRAP residues K757 and F759 are positioned to interact with the 5′ DNA primer, contributing to the "anchor site." K749 and R756 potentially contact the templating region on TR. D788 is a critical contact in the TEN–(IFD-TRAP) interface required to maintain telomerase activity.

more subtle in this case. Despite containing genomes of similar size, *S. cerevisiae* (~12 Mb genome) has 16 chromosomes (Goffeau et al, 1996), whereas *S. pombe* (~14 Mb genome) has only three (Wood et al, 2002). Consistent with our model that invokes the need to enforce telomere specificity in the presence of limiting telomere concentration, *S. pombe* shelterin protein Tpz1, like mammalian TPP1, recruits telomerase to telomeres with the help of a TEL patch–like region in its OB domain (Hu et al, 2016). In contrast, *S. cerevisiae* Est3 (homologous to TPP1 OB) is part of the telomerase holoenzyme and not a telomeric protein (Rao et al, 2014). A more detailed comparison of the genome size, chromosome number, and presence/absence of Est3 in yeast species and other eukaryotes will be needed to test the robustness of our model. But if this correlation is robust, it will imply that when the telomere-number:genome-size ratio is large, the TEN–(IFD-TRAP)–TPP1 structural interface continues to fortify telomerase but may no longer be used as a quality control step to ensure recruitment of the enzyme to telomeres.

Together, our results provide new insights into the functional importance of several TERT (and TPP1) structural elements (Fig 6). The IFD-TRAP of TERT stands out as a critical hub for several macromolecular interactions. As shown previously (Tesmer et al, 2019; Liu et al, 2022; Sekne et al, 2022) and reinforced in our study (by TERT D788), the IFD-TRAP binds the TEN and TPP1 OB domains to form the telomerase–telomere recruitment interface (Fig 6). In this regard, a recent study revealed that point mutations introduced in an isolated IFD-TRAP domain could reduce co-immunoprecipitation with hTPP1 in human cell lysates (Lambert-Lanteigne et al, 2023). Our analysis helps complete the understanding of how TPP1 recruits telomerase by revealing the importance of a hydrophobic residue (F172 and L84 in human and mouse TPP1, respectively) within the acidic TEL patch loop and providing an explanation for the unusual structural basis of TEL patch recognition by TERT (using helical dipole interactions). Our studies also provide credence to the hypothesis that telomerase recruitment to telomeres may extend beyond TPP1

to also include POT1. We highlighted the importance of a region in the TEN domain ($^{99}$FGF$^{101}$ loop) that is closer to POT1-DNA than the TEN–(IFD-TRAP)–TPP1 interface in the cryo-EM structure but is indispensable for telomerase recruitment to telomeres (Fig 6). Future studies will test the molecular basis for its function and potentially reveal how novel TERT interactors (like POT1) assist with holoenzyme recruitment. The function of the IFD-TRAP transcends the formation of the three-way TEN–(IFD-TRAP)–TPP1 interface as we were able to isolate separation-of-function IFD-TRAP mutations that impact telomerase catalysis but not its recruitment to telomeres. Given their location in the cryo-EM structures, TERT K757 and F759 likely form interactions with DNA and TERT K749 and R756 bind TR (Fig 6). Finally, we further characterized motif 3N of human TERT, which seems to dampen telomerase activity. Detailed mechanistic studies are required to understand the payoff for harboring this apparently autoinhibitory region within TERT (Fig 6). In summary, our work provides a biochemical complement to various structural features observed in the cryo-EM structure of TPP1-bound telomerase.

## Materials and Methods

### Molecular cloning and site-directed mutagenesis

The pFBHTb-Smt3star-TPP1-N and pFBHTb-Smt3star-hPOT1 plasmids for coexpressing human TPP1-N and POT1 in High Five insect cells, the pET-Smt3-OB-DBD plasmids for the bacterial expression of fusions of human TPP1 OB domain (aa 88–250) or mouse TPP1 OB domain (aa 1–162) to the human POT1 DBD (aa 2–303) using a glycine-serine linker, and the pVan-3x-Flag-TERTcDNA/6xMyc-His and pVan-3x-Flag-TERTΔTENcDNA/6xMyc-His plasmids for FLAG-tagged TERT expression in HEK 293T cells have been described previously (Tesmer et al, 2019). pcDNA3-TPP1-6xMyc was engineered by cloning TPP1 (aa 87–544) into pcDNA3 using primers containing

sites for restriction enzymes Not1 and Xho1. hTEN (aa 2–197) was cloned into pcDNA3 using restriction enzyme-mediated cloning with primers containing BamH1 and EcoR1 sites. Mutations were introduced into the above plasmids with Phusion polymerase (NEB) and complementary primers containing the desired mutations. IFD-TRAP deletion (Δ735–799) and Motif 3N deletion (Δ641–651) were cloned by PCR amplification of the pVan-3x-Flag-TERTcDNA/6xMyc-His plasmid using 5'-phosphorylated primers that generated a linear PCR product of the plasmid sequence lacking the respective coding sequences but, including a GSG or GSGGSG linker, respectively, in its place. The linear PCR product was circularized with a T4 rapid DNA ligation kit (Thermo Fisher Scientific) to generate the TERT deletion constructs. The gene constructs in all newly cloned plasmids were sequenced to confirm the accuracy of the sequence, including the presence of any intended mutation and the absence of unwanted changes.

## Protein expression and purification

The detailed methods for purifying POT1, TPP1-N, and OB-DBD proteins have been described previously (Grill et al, 2018). Briefly, appropriate isopropyl $\beta$-d-thiogalactopyranoside-induced BL21(DE3) *E. coli* cells or baculovirus-infected High Five insect cells were lysed using sonication. Lysates were subjected to nickel-agarose (Ni-NTA) affinity chromatography and eluted proteins were treated with Ulp1 protease (material transfer agreement with Cornell University for pUlp1 vector) to cleave the Smt3 tag (for proteins expressed in *E. coli*) or with SUMOstar protease (Life Sensors; for proteins expressed insect cells) to cleave the SUMOstar tag. Size exclusion chromatography (Superdex 75 for TPP1-N and Superdex 200 for POT1 and OB-DBD; GE Life Sciences) was used to further purify the proteins. The POT1 protein purification also involved an anion exchange (HiTrap Q HP; GE Life Sciences) step.

## Immunoblotting

The following antibodies were used for immunoblotting: mouse monoclonal anti-FLAG M2-HRP conjugate (A8592; 1:10,000; Sigma-Aldrich), mouse monoclonal anti-c-Myc antibody HRP conjugate (sc-40 HRP; 1:2,000; Santa Cruz), mouse monoclonal anti-$\beta$-actin antibody (A5441; 1:10,000; Sigma-Aldrich), secondary horseradish peroxidase-conjugated goat antibodies against mouse IgG (1:10,000; Santa Cruz Biotechnology). Primary antibodies were visualized by chemiluminescence detection using ECL plus reagents (Pierce ECL Western Blotting Substrate; Thermo Fisher Scientific). The data were visualized using a gel-documentation system (Odyssey Fc; LiCor).

## Cell culture

HeLa-EM2-11ht cells (Dalby et al, 2015) and HEK 293T cells were cultured in DMEM (Gibco) containing 10% FBS and 1% penicillin–streptomycin (Gibco), at 5% $CO_2$, at 37°C.

## IF and FISH microscopy

IF-FISH to measure telomerase recruitment to telomeres was performed as reported previously (Tesmer et al, 2019) with a few modifications. HeLa-EMT-11ht in a six-well plate was transfected with 1 $\mu$g of p3X-Flag-TERT-cDNA-6xMyc-His (WT, deletion variants, or point mutations) and 3 $\mu$g of hTR-Bluescript II SK(+) (and 1 $\mu$g of pCDNA3-6xMyc-hTEN for *trans* complementation experiments, and 500 ng of pCDNA3-TPP1-6xMyc if added) using Lipofectamine LTX, passaged onto coverslips the next day, and fixed at ~48 h after transfection. 1X PBS containing 4% formaldehyde was used to fix the cells for 10 min and 0.3% Triton X-100 1X PBS was used to permeabilize the cells for 5 min, with 1X PBS washes in between. For IF, the cells were first blocked in 1X PBS containing 0.1% Triton X-100 and 3% BSA for 30 min, followed by incubation with Anti-FLAG M2 (1:500; Sigma-Aldrich) antibody in blocking buffer for 1 h. After washing the coverslips three times with 1X PBS, they were incubated with anti-mouse IgG conjugated to Alexa-Fluor 488 (1:500; Life technologies) for 30 min. The cells were washed three times with 1X PBS and fixed again with 4% formaldehyde. Next, the coverslips were washed with 1X PBS three times, dehydrated successively with 70%, 95% and 100% ethanol, air-dried, and rehydrated in 2X SSC-50% formamide for 5 min. The cells were next prehybridized at 37°C for 1 h with hybridization buffer containing 100 mg/ml dextran sulfate, 0.125 mg/ml yeast tRNA, 1 mg/ml BSA, 0.5 mg/ml salmon sperm DNA, 1 mM vanadyl ribonucleoside complexes, and 50% formamide in 2X SSC. The cells were then hybridized with a mix of three cy5-conjugated probes against hTR (30 ng per probe per coverslip) and cy3-conjugated PNA probe (1:1,000) against telomeric DNA overnight at 37°C. The next morning, the coverslips were washed twice with 2X SSC-50% form-amide for 30 min and twice with PBS before being mounted on microscope slides with ProLong Gold mounting medium containing DAPI. All imaging was done on the Leica SP5 laser scanning confocal microscope, using a 100× oil objective. Images were processed with ImageJ and colocalizations were done manually. Telomerase assembly was quantified by counting the number of IF FLAG (TERT variants) foci that were colocalized with hTR FISH foci. Telomere recruitment was quantified by looking for the percentage of assembled telomerase at telomeres. At least 250 TERT and TR foci each were counted for IF-FISH experiments unless specified otherwise.

## Telomerase extract preparation for direct primer extension assay

Telomerase assays (including *trans* complementation) were performed as described previously (Tesmer et al, 2019). 80% confluent HEK 293T cells in a six-well plate were transfected with 1 $\mu$g of 6xMyc-TEN (if included), 1 $\mu$g 3xFlag-TERT variants, and 3 $\mu$g hTR. The cells were split into 6 cm plates the next day and harvested at ~48 h post-transfection. The cells were scraped, washed with 1X PBS, and lysed in 300 $\mu$l CHAPS buffer (10 mM Tris–HCl [pH 7.5], 1 mM $MgCl_2$, 1 mM EGTA, 0.5% CHAPS, 10% glycerol, 5 mM $\beta$-mercaptoethanol), 1X complete Mini EDTA-free Protease Inhibitor Cocktail (11836170001; Roche), and 2.5 $\mu$l RNasin 20 U/$\mu$l (Promega). Cells were rocked at 4°C for 20 min and centrifuged at 16,000$g$ for 10 min, at 4°C. The supernatant was flash-frozen in single-use aliquots, which were thawed immediately before use in telomerase assays.

## Telomerase direct primer extension assay

Primer extension reactions were conducted at 30°C for 1 h, unless specified otherwise, with 4 $\mu$l of super telomerase extracts in a

buffer containing 50 mM Tris–HCl (pH 8.0), 1 mM MgCl$_2$, 1 mM spermidine, 30 mM KCl, 5 mM $\beta$-mercaptoethanol, 1 $\mu$M of primer a5 (TTAGGGTTAGCGTTAGGG), 500 $\mu$M dATP, 500 $\mu$M dTTP, 2.92 $\mu$M un-labeled dGTP, and 0.17 $\mu$M $\alpha$-$^{32}$P-dGTP (3,000 Ci/mmol). POT1–TPP1-mediated processivity stimulation was tested by supplementing reactions with 500 nM of POT1 and TPP1-N or indicated OB-DBD constructs; control samples were supplemented with a protein buffer. Extension reactions were quenched with 100 $\mu$l of a buffer containing 3.6 M ammonium acetate and 20 $\mu$g of glycogen, and the DNA products were precipitated with ethanol at –80°C overnight. The pellets were resuspended in 8 $\mu$l H$_2$O, mixed with 8 $\mu$l loading buffer containing 95% formamide, and heated at 95°C for 5 min. Samples were resolved on a 10% acrylamide, 7 M urea, 1X TBE sequencing-size gel, which was then dried and imaged on a phosphorimager screen (Amersham Typhoon; GE). All assays were analyzed using Imagequant TL (GE Life Sciences) software. Processivity was calculated as previously described (Latrick & Cech, 2010) as the fraction of intensity of repeats is longer than 16 hexad repeats over the intensity of repeats in the entire lane unless specified otherwise. Activity was calculated as the intensity of all the elongation products in the lane normalized against the intensity of all the elongation products in the lane with WT telomerase construct from the same gel.

### Structural comparisons and depictions

Structural alignments were done in either PyMol (DeLano, 1998) or Coot (Emsley & Cowtan, 2004). All structural images were generated in PyMol.

# Data Availability

This study includes no data deposited in external repositories.

# Supplementary Information

# Acknowledgements

We thank members of Nandakumar laboratory for critical feedback on the manuscript. We thank Gregg Sobocinski for help with microscopy. This work was supported by NIH Grants R01GM120094 (J Nandakumar), R01HD108809 (J Nandakumar), R35GM148276 (J Nandakumar), and an American Cancer Society Research Scholar grant RSG-17-037-01-DMC (J Nandakumar).

## Author Contributions

S Padmanaban: conceptualization, formal analysis, validation, investigation, visualization, methodology, and writing—original draft, review, and editing.
VM Tesmer: conceptualization, formal analysis, validation, investigation, visualization, methodology, and writing—original draft, review, and editing.
J Nandakumar: conceptualization, resources, supervision, funding acquisition, validation, visualization, project administration, and writing—review and editing.

## Conflict of Interest Statement

The authors declare that they have no conflict of interest.

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
