## [Reviewer comments · Life Science Alliance]

Life Science Alliance

Interaction hub critical for telomerase recruitment and primer-template handling for catalysis

Shilpa Padmanaban, Valerie Tesmer, and Jayakrishnan Nandakumar

DOI: <https://doi.org/10.26508/lsa.202201727>

Corresponding author(s): Jayakrishnan Nandakumar, University of Michigan-Ann Arbor

Review Timeline:

Submission Date:	2022-09-16
Editorial Decision:	2022-10-21
Revision Received:	2023-02-14
Editorial Decision:	2023-03-08
Revision Received:	2023-03-14
Accepted:	2023-03-15

Scientific Editor: Novella Guidi

Transaction Report:

October 21, 2022

Re: Life Science Alliance manuscript #LSA-2022-01727

Dr. Jayakrishnan Nandakumar
University of Michigan-Ann Arbor
MCDB
1105 N. University avenue
Ann Arbor, MI 48109

Dear Dr. Nandakumar,

Thank you for submitting your manuscript entitled "The insertion in fingers domain of human telomerase is an interaction hub for telomerase function" to Life Science Alliance. The manuscript was assessed by expert reviewers, whose comments are appended to this letter. We invite you to submit a revised manuscript addressing the Reviewer comments.

Thank you for this interesting contribution to Life Science Alliance. We are looking forward to receiving your revised manuscript.

Sincerely,

B. MANUSCRIPT ORGANIZATION AND FORMATTING:

Reviewer #1 (Comments to the Authors (Required)):

Padmanaban et al addresses the structure-function of an intriguing 4-way interface that physically connects the catalytic enzyme of telomerase (TEN and IFD domains of TERT) and its noncoding RNA subunit (TR), with the telomere-capping shelterin complex (TPP1) and telomeric DNA. The formation of the TERT/TR/TPP1 interacting hub at telomeric ends not only represents a defining feature of telomerase RNP but also poses a key rate-limiting step for maintaining chromosomal stability. With the recent breakouts of CryoEM studies spotlighting telomerase and shelterin, there comes a wealth of structural models together with an unmet appetite for much-needed functional testings and validations. Therefore, this work is likely to be of interest to LSA readers in the field of structural, cellular, biochemical, and RNP biology. The experimental results are well-presented, and the manuscript is clearly written.

Meanwhile, the manuscript is confirmatory at large, building upon the framework from published CryoEM studies. In addition, the key concept of TEN domain complementation in trans originated from previous work from Kathleen Collins lab (Wu et al.). Furthermore, the flow of the manuscript lacks a general theme and conclusion, giving out an impression of multiple parallel mini-pieces engrafted or piled onto each other. Thus, instead of concluding on the claim that "using orthologous ways to study structure-function is important", the authors should bridge individual findings in a synthesized summary / central theme. A final figure with a table and/or a TERT+TPP1 domain map should be included to summarize the contributions of various structural features to these complementary readouts in vitro and in cells. Other specific comments are below:

1. Given the focus on the IFD domain and the demonstrated ability to produce TERT Δ IFD and TERT Δ TEN Δ IFD, the degree to which the reconstituted TERT/TPP1/TEN subassembly depends on IFD should be addressed.
2. The authors showed in Fig 3C that TR's presence seems to stabilize interactions between TERT-ring and TEN. TR's contribution to the assembly of TERT/TPP1/TEN tertiary complex should also be addressed. In addition, TR's presence in various immunoprecipitated assemblies should be probed.
3. The higher +1 chain product with no detectable type-II processivity is clearly demonstrated in Δ IFD constructs (Fig 1D), which does not seem to be reproduced in any of the single IFD mutants in Fig 3A. An explanation should be given.
4. The authors demonstrated that Myc-TEN could partially rescue the telomeric targeting defect by FLAG-TERT Δ TEN to about 35% of telomeres. Can a co-IF be performed to also visualize the presence of Myc-TEN at telomeres? Double staining of both FLAG and Myc can further substantiate the notion that the reconstituted licensing to telomeres can be achieved by TEN and TERT-ring trans-complementation.
5. How different TERT variant RNPs are programmed or normalized into each direct assay reaction should be explained in the method session.
6. The gel in Fig 3D showed a seemingly bigger activity differential (summing up all the chain intensity in lane 2 vs. 5 by eyeball test) than the one quantitated in Fig 3E (1~3 fold). A representative gel image should be used.

Reviewer #2 (Comments to the Authors (Required)):

The reviewed manuscript by Padmanaban, Tesmer and Nandakumar, investigated the functional interactions taking place at the interphase between the TERT insertion in fingers domain and TEN domain, the shelterin TPP1-POT1 heterodimer (particularly the TPP1 NOB and TEL patch motifs), the telomerase RNA template and the telomere substrate. The manuscript elucidates this interphase as a critical hub governing telomerase recruitment to telomeres, and its overall activity and repeat addition processivity. Any perturbation to these interactions is expected to affect telomerase action and telomere maintenance. Furthermore, the authors generate separation of function mutations and distinguish the functional importance of the TERT motif 3N, TEN domain loop at positions 99-101, and TEL patch residue F172. They build on accumulating data from Nandakumar, Cech, Collins, Autexier and other labs, not only reinforcing previous hypotheses but also revealing new functions and mechanistic insights. Finally, the authors propose an interesting hypothesis, according to which the absolute requirement of the telomere protein TPP1 for telomerase action provides specificity of telomerase to native telomeres. Furthermore, perhaps these interactions provide a signaling pathway for telomeres to regulate telomerase in cis. Overall, the experiments are most elegant, conducted at the highest scientific standards, and include the proper controls. All conclusions are strongly supported by data, and interpretations or speculations are presented as such. The results beautifully complement and add functionality and mechanistic explanations to recent cryo EM structures reported by the groups of Nguyen, Feigon and Collins. The manuscript is clearly written and, to the best of my evaluation, upon addressing the following comments it is well appropriate for publication at Life Science Alliance.

Specific comments:

1. Introduction, line 5: For the sake of accuracy, I would add that nuclease action is also contributes to telomere shortening. Also, not all dividing cells express telomerase, as may be understood from this sentence.
2. Fig. 1A: The figure indicates IFDa,c, while the text mentions IFDa,b. The structure model is not very clear and hard to identify the elements mentioned in the text (e.g., template-primer duplex). I suggest placing an inset within a complete model and making an effort to make it more accessible to non-structural biologists. This is true also for the structure models in other figures. It is not easy to portray such complex structures in a clear way in 2D, but I think it is worth making a better effort since visual understanding of the structures is so important to help the reader understand the results and see the relation between structure and function.
3. Page 5, first paragraph: Can you indicate these elements on the structure?
4. Second chapter of the results: TERT Δ IFD was not recruited to telomeres. Do the authors know if it interacts with TPP1?
5. Page 12, last paragraph: The first sentence is long. I suggest breaking it in two.
6. Page 13, last paragraph, second sentence: Something doesn't read well. May be remove 'but'?
7. Page 14, first paragraph: when mentioning OB-DBD for the first time, please indicate which part comes from which protein. "the lack of a statistically significant increase in processivity in the presence..." Is there statistically significant decrease from the WT?
8. Page 14, second paragraph: In Fig 4A, by eye, F99V does appear to have lower basal activity. The recruitment results are very interesting. Why not placing Fig. S2E in Fig. 4?
9. Page 16, second paragraph. Looking at the gel image (5F) by eye, the differences in processivity appear more obvious to me than reflected in the graph (5H). My feeling is that the way quantified is not making justice with the results... Please check. Same for Fig. S3B and D
10. Do the authors know or perhaps can speculate in the discussion what is the effect of a mouse TPP1-L84F on the mouse telomerase?
11. Discussion, end of second chapter: Not sure I understand the last speculation. Fidelity was not examined. May be dampening the activity enhances translocation? i.e., there is a trade off between overall activity and RAP here?
12. Discussion, last chapter. The hypothesis about TPP1 as a telomeric protein required for telomerase action, does ensuring specificity to native telomeres is very interesting. I also understand the rationale presented for the ciliate p50 being part of the telomerase complex, since ciliates have thousands of telomeres and also require de novo telomere addition upon the generation of the macronucleus. But I doubt the 16 chromosomes of cerevisiae fall into the same logic. Yeasts do need telomerase to act specifically at native telomeres and not at broken ends. And other budding yeasts with Est3 have less than 16 chromosomes (e.g., 6 in *K. lactis*).

Reviewer #3 (Comments to the Authors (Required)):

The manuscript by Padmanaban and colleagues reports on three-way TEN-IFD-TPP1 interactions important for telomerase recruitment to telomeres and processive telomere repeat addition in the context of recently published cryo-EM structures. They implicate additional motifs and residues in TERT with previously unanticipated roles in telomerase function and a residue in TPP1 that help explain the human and mouse differences in telomerase processivity.

In Figure 1, IP lanes 4 and 5, FLAG-TERT Δ TEN pulls down TPP1, consistent with the TPP1-binding surface of TERT extending beyond TEN and implicates the IFD-TRAP. Does the FLAG-TERT Δ TEN Δ IFD pull down TPP1?

In Figure 2, the authors conclude that the interaction between TEN, IFD and TPP1 in trans drives telomerase recruitment to telomeres highlighting the robustness of the three-way interaction in the cell. Can the observed rescue be improved by addition of exogenous TPP1/Pot1 or would help stimulate telomere localization of telomerase for the FLAG-TERT Δ TEN Δ IFD + TEN?

In Figure 3, mutations in the context of the trans complementation system are introduced to potentially separate phenotypes arising from defective TEN-TRAP-nucleic acid interaction versus TERT-TPP1 interaction. But further in that section, the authors indicate 'As the mutations were targeted to impact IFD binding to TEN domain, TR, or DNA, we expected abrogated trans complementation phenotypes even in the absence of TPP1. But clearly the results show that there is not complete abrogation.

In Figure 3/S1B, D788R appears to be less expressed than WT (also K749E). The authors also suggest that the defect for D788R may be due to the location of D788 adjacent to the N-terminal helix of the TEN domain, causing a weakened TEN-IFD interface. Can the authors introduce this mutation in FLAG-TERT Δ TEN and determine if the interaction with TEN added in trans (+/- TPP1) affects the interaction, or if it affects processivity + TEN in trans +/- TPP1/Pot1 as in Figure 1C and 1D?

Figure 3BC/S1EF, I understand that the variability in the assay makes quantification difficult, but I don't understand the quantification provided in Figure 3B with respect to the numbers cited in the text (e.g. 61% relative processivity for H762E). Also for Figure 3EF, is it possible to provide statistics? And for 3F, the longer products in lane 6 compared to lane 3 are not captured by the method of quantification.

In Figure 3, have the authors tested if the FLAG-TERT Δ TEN Δ 641-651 has defects in binding TEN added in trans? The authors conclude that motif 3N is important for telomerase action, but say their results are consistent with the TEN-IFD-TPP1 interface being intact in the loop deletion mutant.

Figure 4, Why were F99 and F101 changed to V? Perhaps because there is variability in the assay, the quantification of stimulation of processivity for F99V and F101V is statistically not significant. But visually the processivity of these enzymes appear to be stimulated by the addition of TPP1/Pot1. The authors suggest POT1 may play a direct role in telomerase recruitment to telomeres. Can F101V be recruited to telomeres? Does overexpression of Pot1 (or Pot1/Tpp1) rescue the recruitment of F99V to the telomeres in Figure 4E, or is F99V defective in co-IP with Pot1?

The authors show that most of the tested IFD mutants are not defective in recruitment to telomeres (though not all were tested, see Figure S1GH). However, the FLAG-TERT Δ IFD is. Can the authors comment on other IFD residues that have been reported by them and others to be important for recruitment. They show that K757 and F759 contribute to anchor site function but that anchoring of telomerase to telomere DNA is separable from the stable recruitment of telomerase to telomeres. Do the authors speculate that these mutants are recruited to telomeres but not active at telomeres? If some residues contribute to anchor site function, are other residues that are important for recruitment acting in concert with K757 and F759?

Should direct primer extension assays include a loading control for quantification?

Blackburn and Greider 1989 should be Greider and Blackburn 1989.

References at top of page 5 are all *T. thermophila* references but should include the human telomerase structure references with TPP1/Pot1.

We thank the Reviewers for carefully reviewing our manuscript and providing excellent feedback for improving it. We have addressed all the comments raised in our manuscript. Below we respond point-by-point to each comment raised by the Reviewers. The reviewer comments are in blue followed by our response starting with ">>>".

Reviewer #1 (Comments to the Authors (Required)):

Padmanaban et al addresses the structure-function of an intriguing 4-way interface that physically connects the catalytic enzyme of telomerase (TEN and IFD domains of TERT) and its noncoding RNA subunit (TR), with the telomere-capping shelterin complex (TPP1) and telomeric DNA. The formation of the TERT/TR/TPP1 interacting hub at telomeric ends not only represents a defining feature of telomerase RNP but also poses a key rate-limiting step for maintaining chromosomal stability. With the recent breakouts of CryoEM studies spotlighting telomerase and shelterin, there comes a wealth of structural models together with an unmet appetite for much-needed functional testings and validations. Therefore, this work is likely to be of interest to LSA readers in the field of structural, cellular, biochemical, and RNP biology. The experimental results are well-presented, and the manuscript is clearly written.

Meanwhile, the manuscript is confirmatory at large, building upon the framework from published CryoEM studies.

>>>We agree that the large volume of biochemical data from several labs combined with the groundbreaking cryo-EM work in the last few years form the foundations of our study. However, we would argue that the cryo-EM work, in addition to validating previous work, revealed several new features (like IFD-TRAP interactions with TR) that have not been validated functionally. Our current study uses functional readouts to test new hypotheses fueled by cryo-EM work.

In addition, the key concept of TEN domain complementation in trans originated from previous work from Kathleen Collins lab (Wu et al.).

>>>We completely agree that the *trans* complementation assay was developed by Dr. Kathleen Collins's lab, and in accordance with this, we cite two seminal papers from the Collins lab alongside the first mention of the *trans* complementation assay (Robart & Collins Mol. Cell 2011 and Eckert & Collins JBC 2012). The key contribution of our study to *trans* complementation is not its development. Rather, our study using telomerase activity assays and IF-FISH methods provides a mechanism (i.e. TEN-IFD interaction) for *how* the *trans* complementation works.

Furthermore, the flow of the manuscript lacks a general theme and conclusion, giving out an impression of multiple parallel mini-pieces engrafted or piled onto each other. Thus, instead of concluding on the claim that "using orthologous ways to study structure-function is important", the authors should bridge individual findings in a synthesized summary / central theme. A final figure with a table and/or a TERT+TPP1 domain map should be included to summarize the contributions of various structural features to these complementary readouts in vitro and in cells.

>>>We agree with this criticism and apologize for not doing a better job of integrating our various pieces of data into a more congruent, central theme. As suggested by the Reviewer, we have now added a figure at the end of the manuscript (copied below) that showcases all the new findings from our study and puts them in the context of the various regions of TERT, TR, TPP1, POT1, and the primer-template complex. As also requested, we have added a "Conclusion" paragraph at the end of the Discussion section integrating the major findings of our study.

Other specific comments are below:

1. Given the focus on the IFD domain and the demonstrated ability to produce TERT Δ IFD and TERT Δ TEN Δ IFD, the degree to which the reconstituted TERT/TPP1/TEN subassembly depends on IFD should be addressed.

>>>We interpret this comment as a request to perform a co-IP of TEN and TPP1 with TERT Δ TEN lacking IFD (TERT Δ TEN Δ IFD). We performed this experiment, and unexpectedly, deleting the IFD (or deleting both the IFD and motif 3N) in the context of TERT Δ TEN, did not impact co-assembly with TEN and TPP1 by co-IP (copied below). These data are completely consistent with a study published very recently in *JBC* (Jan 2023; DOI: 10.1016/j.jbc.2023.102916) from Dr. Chantal Autexier's group (which we now cite and discuss in our revised manuscript). The authors of this study observed that simultaneous deletion of TEN, IFD, and CTE of TERT did not impact the ability of TERT to co-IP with TPP1. This study also reported that simultaneous deletion of IFD and CTE in the context of TERT Δ TEN did not prevent its co-IP with the isolated TEN domain. The authors of this study also tested several TEN and IFD interface mutations but even these were not able to disrupt co-IP when tested in the context of full-length TERT or TERT Δ TEN. To summarize, neither Dr. Autexier's group nor we have been able to disrupt TERT-TPP1 or TEN-TERT Δ TEN binding with point mutations or gross deletion of multiple domains.

Dr. Autexier and co-authors speculate that the reason for the failure of domain deletions and mutations to disrupt co-IP is that TERT regions outside of TEN and IFD (and CTE) are responsible for binding, but the cryo-EM structures suggest that the TEN-IFD-TPP1-OB three-way complex is not part of any other substantial interface in telomerase. Based on this, it is unclear to us whether the "constitutive binding" phenotype is indicative of a new TERT region or a result of non-specific binding in the co-IP. The latter possibility seems plausible in a TERT overexpression setup as subunits (e.g., H/ACA binding proteins) and accessory factors (e.g., HSP90) required for telomerase RNP assembly may be limiting.

In light of these co-IP data, we have decided not to publish the co-IP data until we gain a better grip on their interpretation. Accordingly, Fig. 1C and the associated text from our first submission have been omitted in this revised submission. Note that this doesn't change the impact of the study (only one panel) or prevent dissemination of the co-IP observations (as they can be gleaned from the recent paper from Dr. Autexier's lab). Instead, we focus on the IF-FISH and telomerase assay readouts in this paper (i.e., all remaining figures and panels in the paper), which are less prone to such artifacts as only assembled/functional telomerase is assayed in these experiments (only

assembled TERT will localize in Cajal bodies and telomeres; only assembled telomerase will show primer extension activity). We thank the reviewer for requesting this experiment, as without it, we would not have realized the issues with the interpretations of our solitary co-IP experiment.

B

2. The authors showed in Fig 3C that TR's presence seems to stabilize interactions between TERT-ring and TEN. TR's contribution to the assembly of TERT/TPP1/TEN tertiary complex should also be addressed. In addition, TR's presence in various immunoprecipitated assemblies should be probed. >>>The observation of the positive impact of TR in the Δ TEN TERT-TEN-TPP1 co-IP is reproducible in our hands. However, in light of the new results (from us and the Autexier group), showing retained binding of TPP1 to TERT lacking TEN, IFD, CTE, and motif 3N, we envision the requested experiment is best justified after we have a better grasp on the interpretation of the co-IP data.

3. The higher +1 chain product with no detectable type-II processivity is clearly demonstrated in Δ IFD constructs (Fig 1D), which does not seem to be reproduced in any of the single IFD mutants in Fig 3A. An explanation should be given.

>>>We reason that the severity of the phenotype seen with single mutations is much milder than a complete domain deletion because the former disrupts only a limited number of IFD functions. Based on the cryo-EM structures, the new IFD mutants we identified in this study are most likely impacting the DNA- and RNA-binding activities of IFD, but not the formation of the TEN-IFD-TPP1 interface; an IFD deletion would abrogate all functions. We have included this explanation in our main text as requested by the Reviewer.

4. The authors demonstrated that Myc-TEN could partially rescue the telomeric targeting defect by FLAG-TERT Δ TEN to about 35% of telomeres. Can a co-IF be performed to also visualize the presence of Myc-TEN at telomeres? Double staining of both FLAG and Myc can further substantiate the notion that the reconstituted licensing to telomeres can be achieved by TEN and TERT-ring trans-complementation.

>>>We have performed IF with myc-TEN (see below). The staining for this protein occurs diffusely throughout the nucleus, making it difficult to visualize TEN localization at telomeres (or Cajal bodies). While the fact that the *trans* complementation works in the IF-FISH context implies that TEN is also at telomeres, we are unable to explicitly visualize this using myc-TEN overexpression.

5. How different TERT variant RNPs are programmed or normalized into each direct assay reaction should be explained in the method session.

>>> We apologize for not normalizing our activity data appropriately in the figures of the first submission. We have now normalized the activity of all mutants to the activity of WT (attributed a value of 1) in the same gel for all gels involving TERT mutants. We have also included information about normalization in the methods.

6. The gel in Fig 3D showed a seemingly bigger activity differential (summing up all the chain intensity in lane 2 vs. 5 by eyeball test) than the one quantitated in Fig 3E (1~3 fold). A representative gel image should be used.

>>> We thank the Reviewer for pointing this out. We have replaced the original gel image with one that better reflects the quantitation.

Reviewer #2 (Comments to the Authors (Required)):

The reviewed manuscript by Padmanaban, Tesmer and Nandakumar, investigated the functional interactions taking place at the interphase between the TERT insertion in fingers domain and TEN domain, the shelterin TPP1-POT1 heterodimer (particularly the TPP1 NOB and TEL patch motifs), the telomerase RNA template and the telomere substrate. The manuscript elucidates this interphase as a critical hub governing telomerase recruitment to telomeres, and its overall activity and repeat addition processivity. Any perturbation to these interactions is expected to affect telomerase action and telomere maintenance. Furthermore, the authors generate separation of function mutations and distinguish the functional importance of the TERT motif 3N, TEN domain loop at positions 99-101, and TEL patch residue F172. They build on accumulating data from Nandakumar, Cech, Collins, Autexier and other labs, not only reinforcing previous hypotheses but also revealing new functions and mechanistic insights. Finally, the authors propose an interesting hypothesis, according to which the absolute requirement of the telomere protein TPP1 for telomerase action provides specificity

of telomerase to native telomeres. Furthermore, perhaps these interactions provide a signaling pathway for telomeres to regulate telomerase in cis. Overall, the experiments are most elegant, conducted at the highest scientific standards, and include the proper controls. All conclusions are strongly supported by data, and interpretations or speculations are presented as such. The results beautifully complement and add functionality and mechanistic explanations to recent cryo EM structures reported by the groups of Nguyen, Feigon and Collins. The manuscript is clearly written and, to the best of my evaluation, upon addressing the following comments it is well appropriate for publication at Life Science Alliance.

Specific comments:

1. Introduction, line 5: For the sake of accuracy, I would add that nuclease action is also contributes to telomere shortening. Also, not all dividing cells express telomerase, as may be understood from this sentence.

>>>We thank the reviewer for this comment. We have now mentioned the importance of nucleases and modified our sentence with the word “most” to convey that not all dividing cells express telomerase.

2. Fig. 1A: The figure indicates IFDa,c, while the text mentions IFDa,b. The structure model is not very clear and hard to identify the elements mentioned in the text (e.g., template-primer duplex). I suggest placing an inset within a complete model and making an effort to make it more accessible to non-structural biologists. This is true also for the structure models in other figures. It is not easy to portray such complex structures in a clear way in 2D, but I think it is worth making a better effort since visual understanding of the structures is so important to help the reader understand the results and see the relation between structure and function.

>>>We have corrected IFDa,b in the text to IFDa,c. As suggested by the reviewer, we have now included the full structure and zoomed in on the area of interest for structural depictions in Figures 1 and 4. Due to space limitations in Figure 3 (there are 8 panels), we have provided the full structure and insets, as requested, in the revised supplemental figure Fig. S3.

3. Page 5, first paragraph: Can you indicate these elements on the structure?

>>>We have now indicated the TEL patch and NOB in Figure 1A.

4. Second chapter of the results: TERTΔIFD was not recruited to telomeres. Do the authors know if it interacts with TPP1?

>>> This point was also raised by Reviewer 1, and replicated here is our response to them. co-IP of TEN and TPP1 with TERTΔTEN lacking IFD (TERTΔTENΔIFD): We performed this experiment, and unexpectedly, deleting the IFD (or deleting both the IFD and motif 3N) in the context of TERTΔTEN, did not impact co-assembly with TEN and TPP1 by co-IP (copied below). These data are completely consistent with a study published very recently in *JBC* (Jan 2023; DOI: 10.1016/j.jbc.2023.102916) from Dr. Chantal Autexier's group (which we now cite and discuss in our revised manuscript). The authors of this study observed that simultaneous deletion of TEN, IFD, and CTE of TERT did not impact the ability of TERT to co-IP with TPP1. This study also reported that simultaneous deletion of IFD and CTE in the context of TERTΔTEN did not prevent its co-IP with the isolated TEN domain. The authors of this study also tested several TEN and IFD interface mutations but even these were not able to disrupt co-IP when tested in the context of full-length TERT or TERTΔTEN. To summarize, neither Dr. Autexier's group nor we have been able to disrupt TERT-TPP1 or TEN-TERTΔTEN binding with point mutations or gross deletion of multiple domains.

Dr. Autexier and co-authors speculate that the reason for the failure of domain deletions and mutations to disrupt co-IP is that TERT regions outside of TEN and IFD (and CTE) are responsible for binding, but the cryo-EM structures suggest that the TEN-IFD-TPP1-OB three-way complex is not

part of any other substantial interface in telomerase. Based on this, it is unclear to us whether the “constitutive binding” phenotype is indicative of a new TERT region or a result of non-specific binding in the co-IP. The latter possibility seems plausible in a TERT overexpression setup as subunits (e.g., H/ACA binding proteins) and accessory factors (e.g., HSP90) required for telomerase RNP assembly may be limiting.

In light of these co-IP data, we have decided not to publish the co-IP data until we gain a better grip on their interpretation. Accordingly, Fig. 1C and the associated text from our first submission have been omitted in this revised submission. Note that this doesn’t change the impact of the study (only one panel) or prevent dissemination of the co-IP observations (as they can be gleaned from the recent paper from Dr. Autexier’s lab). Instead, we focus on the IF-FISH and telomerase assay readouts in this paper (i.e., all remaining figures and panels in the paper), which are less prone to such artifacts as only assembled/functional telomerase is assayed in these experiments (only assembled TERT will localize in Cajal bodies and telomeres; only assembled telomerase will show primer extension activity). We thank the reviewer for requesting this experiment, as without it, we would not have realized the issues with the interpretations of our solitary co-IP experiment.

5. Page 12, last paragraph: The first sentence is long. I suggest breaking it in two.

>>>We have broken this sentence into two as requested.

6. Page 13, last paragraph, second sentence: Something doesn't read well. May be remove 'but'?

>>>We agree. We removed the word “but” in this sentence and introduced a “,” to fix the issue.

7. Page 14, first paragraph: when mentioning OB-DBD for the first time, please indicate which part comes from which protein. "the lack of a statistically significant increase in processivity in the presence..." Is there statistically significant decrease from the WT?

>>>We have defined OB-DBD alongside its first mention in the manuscript. We apologize for the confusion regarding the statistical significance of processivity. It arises because of the way we use statistical significance to evaluate the phenotype. We have reported this analysis previously (Tesmer and Smith et al PNAS 2019). The phenotype that is being measured in this experiment is the loss of the stimulation of telomerase processivity seen typically (i.e., for WT telomerase) with POT1-TPP1 (or OB-DBD). For WT telomerase, there is a statistically significant increase in processivity when POT1-TPP1 is added. For F99V telomerase, the increase is not substantial, resulting in a loss of this statistical significance.

8. Page 14, second paragraph: In Fig 4A, by eye, F99V does appear to have lower basal activity. The recruitment results are very interesting. Why not placing Fig. S2E in Fig. 4?

>>>The activity defects were subtle for the two F→V mutants, which is why they were not included (data not shown). We now show quantitation of the recruitment defects in the main figure, as requested.

9. Page 16, second paragraph. Looking at the gel image (5F) by eye, the differences in processivity appear more obvious to me than reflected in the graph (5H). My feeling is that the way quantified is not making justice with the results... Please check. Same for Fig. S3B and D

>>>We agree with the reviewer that the visual defects look more impressive than the quantitation. In our experience, this is routine for telomerase primer extension experiments (i.e., differences are more obvious visually than upon quantitation). We have quantified such data using the method we show as well as the standard method that accounts for the number of G’s (radiolabeled nucleotide) per repeat (data not shown). However, both methods yield similar results and neither matches the severity of the phenotype gleaned by visualization of the gel.

10. Do the authors know or perhaps can speculate in the discussion what is the effect of a mouse

TPP1-L84F on the mouse telomerase?

>>> We only describe assays with human telomerase in this manuscript. But we have performed the requested experiment with mouse telomerase in the context of a larger experiment we performed previously (see below). With telomerase containing mTERT, changing the mouse TPP1 NOB (compare lanes 6 & 7 in panel B), but not the mouse TPP1 L84 (compare lanes 6 & 8 in panel B), to the human TPP1 equivalent showed a reduction in telomerase processivity. Thus, it seems as if human TERT discriminates against mouse TPP1 L84 and NOB, but mouse TERT discriminates against human TPP1 NOB but not TPP1 F172. The structural basis for this is unclear from the human cryo-EM structures and mouse telomerase AlphaFold models that we generated from it (not shown).

11. Discussion, end of second chapter: Not sure I understand the last speculation. Fidelity was not examined. May be dampening the activity enhances translocation? i.e., there is a trade off between overall activity and RAP here?

>>>The Reviewer is correct in that we have not analyzed fidelity. What we observe is that presence of motif 3N reduces activity. Increasing translocation seems like a plausible function and we thank the reviewer for suggesting this. We have included this possibility in our revised manuscript and mention that more experiments need to be performed in the future to fully delineate the role of this TERT element.

12. Discussion, last chapter. The hypothesis about TPP1 as a telomeric protein required for telomerase action, does ensuring specificity to native telomeres is very interesting. I also understand the rationale presented for the ciliate p50 being part of the telomerase complex, since ciliates have thousands of telomeres and also require de novo telomere addition upon the generation

of the macronucleus. But I doubt the 16 chromosomes of cerevisiae fall into the same logic. Yeasts do need telomerase to act specifically at native telomeres and not at broken ends. And other budding yeasts with Est3 have less than 16 chromosomes (e.g., 6 in *K. lactis*).

>>>We agree with the Reviewer that the difference in chromosome number between the two popular yeast models is much smaller than that between humans and ciliates. We have included the following sentences in this section of the Discussion to tone down the generalization of our model. “A similar distinction can be made for the two popular yeast models, *Saccharomyces cerevisiae* (*S. cerevisiae*) and *Schizosaccharomyces pombe* (*S. pombe*), although the difference in telomere number is more subtle in this case. “ And “A more detailed comparison of the genome size, chromosome number, and presence/absence of Est3 in yeast species and other eukaryotes will be needed to test the robustness of our model. But if this correlation is robust, it will imply that when the telomere-number: genome-size ratio is large, the TPP1-TEN-(IFD-TRAP) structural interface continues to fortify telomerase but may no longer be used as a quality control step to ensure recruitment of the enzyme to telomeres.”

Reviewer #3 (Comments to the Authors (Required)):

The manuscript by Padmanaban and colleagues reports on three-way TEN-IFD-TPP1 interactions important for telomerase recruitment to telomeres and processive telomere repeat addition in the context of recently published cryo-EM structures. They implicate additional motifs and residues in TERT with previously unanticipated roles in telomerase function and a residue in TPP1 that help explain the human and mouse differences in telomerase processivity.

In Figure 1, IP lanes 4 and 5, FLAG-TERT Δ TEN pulls down TPP1, consistent with the TPP1-binding surface of TERT extending beyond TEN and implicates the IFD-TRAP. Does the FLAG-TERT Δ TEN Δ IFD pull down TPP1?

>>> This point was also raised by Reviewer 1, and replicated here is our response to them. co-IP of TEN and TPP1 with TERT Δ TEN lacking IFD (TERT Δ TEN Δ IFD): We performed this experiment, and unexpectedly, deleting the IFD (or deleting both the IFD and motif 3N) in the context of TERT Δ TEN, did not impact co-assembly with TEN and TPP1 by co-IP (copied below). These data are completely consistent with a study published very recently in *JBC* (Jan 2023; DOI: 10.1016/j.jbc.2023.102916) from Dr. Chantal Autexier’s group (which we now cite and discuss in our revised manuscript). The authors of this study observed that simultaneous deletion of TEN, IFD, and CTE of TERT did not impact the ability of TERT to co-IP with TPP1. This study also reported that simultaneous deletion of IFD and CTE in the context of TERT Δ TEN did not prevent its co-IP with the isolated TEN domain. The authors of this study also tested several TEN and IFD interface mutations but even these were not able to disrupt co-IP when tested in the context of full-length TERT or TERT Δ TEN. To summarize, neither Dr. Autexier’s group nor we have been able to disrupt TERT-TPP1 or TEN-TERT Δ TEN binding with point mutations or gross deletion of multiple domains.

Dr. Autexier and co-authors speculate that the reason for the failure of domain deletions and mutations to disrupt co-IP is that TERT regions outside of TEN and IFD (and CTE) are responsible for binding, but the cryo-EM structures suggest that the TEN-IFD-TPP1-OB three-way complex is not part of any other substantial interface in telomerase. Based on this, it is unclear to us whether the “constitutive binding” phenotype is indicative of a new TERT region or a result of non-specific binding in the co-IP. The latter possibility seems plausible in a TERT overexpression setup as subunits (e.g., H/ACA binding proteins) and accessory factors (e.g., HSP90) required for telomerase RNP assembly may be limiting.

In light of these co-IP data, we have decided not to publish the co-IP data until we gain a better grip on their interpretation. Accordingly, Fig. 1C and the associated text from our first submission have been omitted in this revised submission. Note that this doesn’t change the impact of the study (only

one panel) or prevent dissemination of the co-IP observations (as they can be gleaned from the recent paper from Dr. Autexier's lab). Instead, we focus on the IF-FISH and telomerase assay readouts in this paper (i.e., all remaining figures and panels in the paper), which are less prone to such artifacts as only assembled/functional telomerase is assayed in these experiments (only assembled TERT will localize in Cajal bodies and telomeres; only assembled telomerase will show primer extension activity). We thank the reviewer for requesting this experiment, as without it, we would not have realized the issues with the interpretations of our solitary co-IP experiment.

In Figure 2, the authors conclude that the interaction between TEN, IFD and TPP1 in trans drives telomerase recruitment to telomeres highlighting the robustness of the three-way interaction in the cell. Can the observed rescue be improved by addition of exogenous TPP1/Pot1 or would help stimulate telomere localization of telomerase for the FLAG-TERT Δ TEN Δ IFD + TEN?

>>>We performed the requested IF-FISH experiment and included it in Fig. S1B-D accompanying our revised manuscript. As you can see in our revised figure (copied below), TPP1 addition does not result in a further increase in the rescue of telomerase recruitment (i.e., it is ~35%). This is not surprising and is fully consistent with our *trans* complementation primer extension results wherein the low signal (activity, not processivity) of *trans* complementation is not fully rescued to full-length TERT levels by the addition of POT1-TPP1. We believe that only a small fraction of the TEN and/or TERT Δ TEN is folded properly in the cell, making one of them the limiting factor and culminating in low telomerase activity. Accordingly, the addition of TPP1 in this context is not able to improve TEN and TERT Δ TEN assembly.

In Figure 3, mutations in the context of the trans complementation system are introduced to potentially separate phenotypes arising from defective TEN-TRAP-nucleic acid interaction versus TERT-TPP1 interaction. But further in that section, the authors indicate 'As the mutations were targeted to impact IFD binding to TEN domain, TR, or DNA, we expected abrogated trans complementation phenotypes even in the absence of TPP1. But clearly the results show that there is not complete abrogation.

>>>We agree that the usage of "abrogate(d)" is not justified in this case, especially since we introduced single (i.e., subtle) mutations within TERT. We have now substituted "abrogated" with "reduced" in this sentence.

In Figure 3/S1B, D788R appears to be less expressed than WT (also K749E). The authors also suggest that the defect for D788R may be due to the location of D788 adjacent to the N-terminal helix of the TEN domain, causing a weakened TEN-IFD interface. Can the authors introduce this mutation in FLAG-TERT Δ TEN and determine if the interaction with TEN added in trans (+/- TPP1) affects the interaction, or if it affects processivity + TEN in trans +/- TPP1/Pot1 as in Figure 1C and 1D?

>>>We did not perform the interaction assay by co-IP due to the caveats of the assay already mentioned above. We wanted to point out that the primer extension results in Figure 3 for D788 were performed with the FLAG-TERT Δ TEN and TEN *trans* complementation constructs.

Figure 3BC/S1EF, I understand that the variability in the assay makes quantification difficult, but I don't understand the quantification provided in Figure 3B with respect to the numbers cited in the text (e.g. 61% relative processivity for H762E). Also for Figure 3EF, is it possible to provide statistics? And for 3F, the longer products in lane 6 compared to lane 3 are not captured by the method of quantification.

>>>We apologize for the confusion regarding H762E. The processivity plot is accurate but the numbers in the text were inaccurate. We have now put the correct percentages for processivity (86% of WT) and activity (68% of WT) in the text.

In Figure 3, have the authors tested if the FLAG-TERT Δ TEN Δ 641-651 has defects in binding TEN added in trans? The authors conclude that motif 3N is important for telomerase action, but say their results are consistent with the TEN-IFD-TPP1 interface being intact in the loop deletion mutant.

>>>As the *trans* complementation primer extension works, it is safe to conclude that the TEN-(IFD-TRAP) interface is intact in this mutant. Furthermore, as POT1-TPP1 increases the production of longer products with this mutant, it is also likely that the entire TEN-(IFD-TRAP)-TPP1 interface is intact despite motif 3N deletion.

We performed a co-IP as implied by the Reviewer's comment, and as already described above (see co-IP data above in response to Reviewer 1's comments), the interpretation of our co-IP assay precludes us from making strong conclusions about the lack of a binding defect we observed for the motif 3N loop deletion mutant and TEN.

Figure 4, Why were F99 and F101 changed to V? Perhaps because there is variability in the assay, the quantification of stimulation of processivity for F99V and F101V is statistically not significant. But visually the processivity of these enzymes appear to be stimulated by the addition of TPP1/Pot1. The authors suggest POT1 may play a direct role in telomerase recruitment to telomeres. Can F101V be recruited to telomeres? Does overexpression of Pot1 (or Pot1/Tpp1) rescue the recruitment of F99V to the telomeres in Figure 4E, or is F99V defective in co-IP with Pot1?

>>>As phenylalanines are hydrophobic residues and at least F101 seems as if it is partly buried in the telomerase structure, we decided to make a more conservative change of F \rightarrow V rather than F \rightarrow A (hydrophobicity of F>V>A) so as to not grossly impact the fold of the TEN domain. As this assay is done with full-length proteins, the signal-to-noise is reasonably high. Therefore, we do not envision the lack of statistical significance coming from more error in the F99V (than WT) experiment. The error bars are also comparable for WT and mutants in our data. We have used this analysis

previously with other mutations in TERT (N125A, T128I; Tesmer and Smith PNAS 2019), which were shown to indeed be TPP1-proximal by the newly solved cryo-EM structures.

We performed telomerase recruitment analysis with F101V and it shows a telomerase recruitment defect (~25% of WT), albeit at a level that is less severe than that of F99V (~0% of WT) (see below). We have now included this data in the revised manuscript. We did not perform co-IP with these TERT mutants due to the caveats of this assay that we described above.

The suggestion to include POT1 in the recruitment assay is excellent. We are very interested in understanding the direct role if any, that POT1 plays in telomerase recruitment. However, we envision this and other experiments directly probing the role of POT1 in telomerase function are beyond the scope of the current study and would constitute a future study.

The authors show that most of the tested IFD mutants are not defective in recruitment to telomeres (though not all were tested, see Figure S1GH). However, the FLAG-TERT Δ IFD is. Can the authors comment on other IFD residues that have been reported by them and others to be important for recruitment. They show that K757 and F759 contribute to anchor site function but that anchoring of telomerase to telomere DNA is separable from the stable recruitment of telomerase to telomeres. Do the authors speculate that these mutants are recruited to telomeres but not active at telomeres? If some residues contribute to anchor site function, are other residues that are important for recruitment acting in concert with K757 and F759?

>>>Yes, our data suggest that K757 and F759 are not involved in TEN-TPP1 binding and are therefore non-essential for telomerase recruitment to telomeres. Yes, functional/mutagenesis data from us and others combined with cryo-EM data suggest that different regions of IFD are dedicated to different aspects of telomerase function, including RNP assembly (TEN-IFD interaction), telomerase recruitment (TEN-IFD-TPP1 interaction), and catalysis (anchoring telomerase to DNA through IFD-DNA interactions and IFD-RNA interactions). The Reviewer asked us to comment on residues important for telomerase recruitment. We have previously reported E793/Q794; The Autexier group has reported (P721, T726, V763, P785, V791, L805, and R811). However, many of these residues were implicated in telomerase recruitment before the solution of the cryo-EM structures. The structures reveal that some of the residues (e.g., P721, T726) are not in the IFD-TRAP but rather in the bracing helices. Similarly, some of the residues (e.g., E793) are binding TEN, not TPP1, while others are likely important for IFD folding (e.g., V791, L805), not directly to its binding to TPP1. Thus,

while these residues when mutated impact telomerase recruitment, many of them may not be directly involved in binding TPP1.

Should direct primer extension assays include a loading control for quantification?

>>>As a bulk of our previous and current work involved quantitation of telomerase processivity, which is mostly insensitive to loading differences (as it is derived by normalizing against the total counts within that lane), we have not routinely used a loading control. We did perform a new telomerase primer extension by including a loading control. However, the low signal of our *trans* complementation was highlighted even in this experiment as the loading control signal was fully saturated at exposures required to visualize *trans* complementation primer extension products (not shown). We argue that the low signal of our *trans* complementation contributes more to the overall error than potential loading issues. Therefore, a visual comparison of the various replicates (done with different telomerase transfections on different days) might offer the best avenue to qualitatively assess activity defects in our *trans* complementation experiments. We have pasted below three different replicates for the Reviewer's perusal to demonstrate that the mutants deemed defective in this study were reproducibly defective across all replicates tested (see below).

Replicate 2

+ WT TEN

TERT Δ TEN
variant
PT

Lanes 1 2 3 4 5 6 7 8 9 10 11 12 13 14 15 16 17 18 19 20

Replicate 3

Blackburn and Greider 1989 should be Greider and Blackburn 1989.
>>> We have corrected this citation in the revised manuscript.

References at top of page 5 are all *T. thermophila* references but should include the human telomerase structure references with TPP1/Pot1.
>>>We have now included the references for the human telomerase +POT1-TPP1 structures.

March 8, 2023

RE: Life Science Alliance Manuscript #LSA-2022-01727R

Dr. Jayakrishnan Nandakumar
University of Michigan-Ann Arbor
MCDB
1105 N. University avenue
Ann Arbor, MI 48109

Dear Dr. Nandakumar,

Thank you for submitting your revised manuscript entitled "Interaction hub critical for telomerase recruitment and primer-template handling for catalysis". We would be happy to publish your paper in Life Science Alliance pending final revisions necessary to meet our formatting guidelines. We have also consulted with an advisor on whether there was a need to perform the co-IP experiment suggested by reviewer 1, and we have agreed that we do not consider that this part of the work is important for the conclusions made in this manuscript.

- please add the weight to the blots in figures S1A, S2A and D, and S3E
- please incorporate your conclusion section into your discussion section

A. FINAL FILES:

B. MANUSCRIPT ORGANIZATION AND FORMATTING:

**Submission of a paper that does not conform to Life Science Alliance guidelines will delay the acceptance of your

manuscript.**

The license to publish form must be signed before your manuscript can be sent to production. A link to the electronic license to publish form will be sent to the corresponding author only. Please take a moment to check your funder requirements.

Sincerely,

Reviewer #1 (Comments to the Authors (Required)):

The reviewer welcomes the new addition of the summary figure in Fig 6. However reviewer regrets author's decision to drop the previous Fig.1C for good, which has been an informative companion with the trans-complementation and recruitment assays. The newly included FLAG-IP "B" figure in the p2p is an informative addition that has a major influence on the overall interpretation, as it suggests the potential presence of additional IFD-independent molecular interface(s) sufficient to stabilize the 3-way hub, but insufficient to support catalysis and recruitment. Since this point was raised by multiple reviewers, the reviewer suggests the author to address the validity of the co-IP directly with known interaction-deficient TPP1 and/or TEN mutants. If those mutants still "stick" to the FLAG-TERT bait with comparable efficiency, then dropping the co-IP results can only become a justifiable decision.

Reviewer #2 (Comments to the Authors (Required)):

In the revised manuscript, the authors adequately addressed all my concerns. They corrected text issues, added structural context to the figures, a new summarizing model and a conclusions paragraph to the discussion. They also removed co-IP experiments, which were hard to interpret and require more work, but were not critical for this manuscript. Altogether, the manuscript increased in clarity, focus and impact, and is ready for publication.

Reviewer #3 (Comments to the Authors (Required)):

I have carefully reviewed the revised manuscript and the response to reviewers. I am satisfied the authors have adequately revised the manuscript and responded to reviewers.

please add the weight to the blots in figures S1A, S2A and D, and S3E

>>>Added

-please incorporate your conclusion section into your discussion section

>>>Removed the word conclusion so that this paragraph is part of Discussion

>>>Not applicable.

>>>We do not intend to do this.

To upload the final version of your manuscript, please log in to your account: <https://lsa.msubmit.net/cgi-bin/main.plex>

A. FINAL FILES:

-- Summary blurb (enter in submission system): A short text summarizing in a single sentence the study (max. 200 characters including spaces). This text is used in conjunction with the titles of papers, hence should be informative and complementary to the title. It should describe the context and significance of the findings for a

general readership; it should be written in the present tense and refer to the work in the third person. Author names should not be mentioned.

B. MANUSCRIPT ORGANIZATION AND FORMATTING:

>>>We have uploaded all these files.

Sincerely,

Reviewer #1 (Comments to the Authors (Required)):

The reviewer welcomes the new addition of the summary figure in Fig 6. However reviewer regrets author's decision to drop the previous Fig.1C for good, which has been an informative companion with the trans-complementation and recruitment assays. The newly included FLAG-IP "B" figure in the p2p is an informative addition that has a major influence on the overall interpretation, as it suggests the potential presence of additional IFD-independent molecular interface(s) sufficient to stabilize the 3-way hub, but insufficient to support catalysis and recruitment. Since this point was raised by multiple reviewers, the reviewer suggests the author to address the validity of the co-IP directly with known interaction-deficient TPP1 and/or TEN mutants. If those mutants still "stick" to the FLAG-TERT bait with comparable efficiency, then dropping the co-IP results can only become a justifiable decision.

>>>The Editor has already addressed this point.

Reviewer #2 (Comments to the Authors (Required)):

In the revised manuscript, the authors adequately addressed all my concerns. They corrected text issues, added structural context to the figures, a new summarizing model and a conclusions paragraph to the discussion. They also removed co-IP experiments, which were hard to interpret and require more work, but were not critical for this manuscript. Altogether, the manuscript increased in clarity, focus and impact, and is ready for publication.

>>>Thank you.

Reviewer #3 (Comments to the Authors (Required)):

I have carefully reviewed the revised manuscript and the response to reviewers. I am satisfied the authors have adequately revised the manuscript and responded to reviewers.

>>>Thank you.

March 15, 2023

RE: Life Science Alliance Manuscript #LSA-2022-01727RR

Dr. Jayakrishnan Nandakumar
University of Michigan-Ann Arbor
MCDB
1105 N. University avenue
Ann Arbor, MI 48109

Dear Dr. Nandakumar,

Thank you for submitting your Research Article entitled "Interaction hub critical for telomerase recruitment and primer-template handling for catalysis". It is a pleasure to let you know that your manuscript is now accepted for publication in Life Science Alliance. Congratulations on this interesting work.

DISTRIBUTION OF MATERIALS:

Again, congratulations on a very nice paper. I hope you found the review process to be constructive and are pleased with how the manuscript was handled editorially. We look forward to future exciting submissions from your lab.

Sincerely,
